# Structure of the MUC5AC VWD3 assembly responsible for the formation of net-like mucin polymers

Sergio Trillo-Muyo [ORCID] ✉, Anna Ermund [ORCID] & Gunnar C Hansson [ORCID] ✉

## Abstract

Gel-forming mucins MUC5AC and MUC5B constitute the main structural component of the mucus in the respiratory system. Secreted mucins interact specifically with each other and other molecules giving mucus specific properties. We determined the cryoEM structures of the wild type D3 assembly of the human MUC5AC mucin and the structural single nucleotide polymorphisms (SNP) variants Arg996Gln and Arg1201Trp that affect intermolecular interactions. Our structures explain the MUC5AC N-terminal non-covalent oligomerization after secretion. The D3 assembly forms covalent dimers that can appear in two alternative conformations, open and closed, where the closed conformation dimers interact through an arginine-rich loop in the TIL3 domain to form tetramers. Our study provides a model to explain MUC5AC net-like structures and how the two SNPs will affect mucus organization, something that might affect lung and other diseases.

Keywords Mucin; Stomach; Respiratory Tract; Cryo-electron Microscopy; Von Willebrand D Domain
Subject Categories Molecular Biology of Disease; Respiratory System; Structural Biology

## Introduction

Mucins are a group of highly glycosylated molecules that cover and protect all mucosal surfaces of the body (Hansson, 2020). The two gel-forming mucins MUC5AC and MUC5B constitute the main structural components of the mucus protecting the underlying epithelia in the respiratory system. Both mucins are large proteins, 5654 or 5762 amino acids. All the gel-forming mucins and the von Willebrand factor (VWF) have an N-terminal part built by 3.5 von Willebrand D assemblies, VWD1, VWD2, VWD', and VWD3. The VWD assemblies are formed by the VWD domain, C8, trypsin inhibitor like (TIL) and E domains, except the VWD' domain that only has the TIL and E domains. These mucins have their N-termini followed by one or several PTS domains rich in proline, threonine, and serine. These hydroxy amino acids become heavily O-glycosylated to form the extended rod-like mucin domains. The dense sugar coating is responsible for the mucin hygroscopicity necessary for binding water and to mucus formation. The PTS domains are interrupted by CysD domains, nine in MUC5AC and seven in MUC5B, involved in homotypic interactions as shown for CysD2 of MUC2 (Recktenwald et al, 2024). The MUC5AC, MUC5B, and MUC2 mucin C-terminal region is formed by a VWD4 assembly, 3.5 VWC domains and a C-terminal cysteine-knot (Gallego et al, 2023).

Secreted mucins interact specifically with each other and other molecules to give mucus its specific properties. The mucins are orderly packed in the goblet cell granule due to low pH and high $Ca^{2+}$. Upon secretion, they are unpacked into large disulfide-bond polymers, a process that requires an increased pH by bicarbonate transported via the cystic fibrosis transmembrane conductance regulator (CFTR). The intermolecular disulfide bonds in the ER and trans-Golgi network in the C-terminus and the N-terminal regions allow the expansion into covalent linear oligomers. However, additional non-covalent interactions are required for mucus organization.

The respiratory system is constantly exposed to inhaled particles, bacteria and viruses. Humans have submucosal glands down to the 10th bronchial bifurcation where chloride and bicarbonate-rich fluid pulls out the MUC5B mucin into long polymers exiting the glands as >20 µm thick bundled strands (Ermund et al, 2017; Hoegger et al, 2014; Widdicombe and Wine, 2015). These are patchily coated with the MUC5AC from the surface goblet cells which controls the bundle movement by attachment/detachment events (Bos et al, 2023; Ermund et al, 2018; Ermund et al, 2017). The MUC5AC and MUC5B mucins together clean the larger airways.

The MUC5B mucin is required for the normal lung homeostasis in mice whereas the MUC5AC is not (Roy et al, 2014). Interestingly, the MUC5AC mucin is increased in amounts at lung diseases (Fahy and Dickey, 2010; Fernandez-Blanco et al, 2018) where the surface goblet cells co-express both MUC5B and MUC5AC in the same cell (Hoang et al, 2022), leading to the formation of an attached stratified mucus layer (Fernandez-Blanco et al, 2018).

Department of Medical Biochemistry and Cell Biology, University of Gothenburg, 40530 Gothenburg, Sweden. ✉E-mail: sergio.muyo@medkem.gu.se; gunnar.hansson@medkem.gu.se

The question why we have two different mucins in the lung has been a puzzle. The MUC5B mucin is clearly forming linear molecules as confirmed by electron microscopy (Ermund et al, 2017; Ridley et al, 2016). The MUC5AC mucin on the other hand forms net-like structures (Carpenter et al, 2021). To address this question, we have performed structural studies using cryoEM. The results show that the disulfide bonded linear MUC5AC mucin also interacts non-covalently in its VWD3 assembly where there are genetic variants that are affecting the tendency to form non-covalent tetra- or oligomers.

## Results and discussion

### MUC5AC-D3 covalent dimer

Expression of the secreted complete N-terminal part of MUC5AC (MUC5AC-N) has proved to be challenging and has only been achieved at low yields in polarized airway cell lines (Ryan et al, 2015). We have also tried to produce the complete secreted MUC5AC in CHO and HEK cells without any success. The VWD1-VWD2 (D1-D2) assemblies of VWF and MUC2 are cleaved off after polymerization in the late Golgi suggesting that these domains are primary required for the granule packing (Sadler, 1998; Schutte et al, 2014). MUC2 D1-D2 assemblies are compactly packed at low pH in the goblet cell granule at the same time as they are involved in intracellular polymerization (Javitt et al, 2020). MUC2-N expressed in CHO cells and analyzed by cryoEM at pH 7.4 shows that only the compact D3 dimer assembly can be visualized as the D1, D2, and D' likely are too flexible. This suggests less importance of D1-D2 for the mature secreted mucins at physiological, neutral pH. Unable to produce the mature, secreted MUC5AC-N, we had to focus on the VWD3 assembly, the MUC5AC-D3 assembly and its interactions (Fig. 1A,B).

We designed expression plasmids including the N-terminal D' assembly, the D3 assembly and the CysD1 domain with and without an N-terminal 6xHis-tag (Fig. 1B). These plasmids were expressed in Lec 3.2.8.1 CHO cells and the recombinant proteins purified and analyzed by gel electrophoresis showing the expected band sizes after reduction (Fig. 1C). Without reduction, the bands migrate approximately at the double size, suggesting that all three recombinant proteins are disulfide-bonded dimers.

The structure of the MUC5AC-D3 assembly was analyzed by cryoEM at pH 7.4 (Appendix Table S1, Appendix Fig. S1). The cryoEM reconstruction of the D'-D3-CysD1 recombinant protein reached higher resolution (3.2 Å) than the isolated D3, even if only the D3 assembly is visible due to flexibility of the D' and CysD1 domains (Fig. 1D). The overall structure of the MUC5AC-D3 assembly dimer is similar to the previously reported D3 assemblies (Dong et al, 2019; Javitt et al, 2019). The covalent MUC5AC-D3 assembly dimer is formed via two intermolecular disulfide bonds, Cys1132-Cys1132 in the C8-3 domain and Cys1174-Cys1174 in the TIL3 domain (Fig. 1D–F). However, the Cys1132 intermolecular disulfide bond is partially reduced as observed previously in MUC2-D3 (Javitt et al, 2019). The interaction surface is highly conserved between MUC5AC-D3 and MUC2-D3 with only three relevant substitutions. In the C8-3 domain interaction region MUC5AC-D3 presents a phenylalanine (Phe1086, Fig. 1E) instead of a histidine (MUC2-D3 His1042), potentially affecting the effect of pH in the intracellular packing. In the

TIL3 domain interface, Ser1154 (Fig. 1F) substitutes a phenylalanine present in an equivalent position in MUC2 (Phe1110). This serine is highly conserved in MUC5AC and MUC5B between species. The hydrophobic pocket formed by Phe1164', Tyr1167', and Tyr1168' interacts with Pro1158, while Tyr1178, Leu1163 and the aliphatic chain of Arg1156 creates another pocket that accommodates Phe1164' (Fig. 1F). In MUC2 the second hydrophobic pocket is also formed by Phe1110 stabilizing the interaction with Phe1164'. In the same area, MUC2 interfacing residues Arg973'-Asp1115 are substituted in MUC5AC by Ser1159-Lys1017' resulting in a loss of a salt bridge. No hydrogen bonds between Ser1159-Lys1017' were observed.

The structure of the MUC2-E3 domain have been solved by crystallography due to the crystal packaging contacts showing high B-factors (Javitt et al, 2019). However, in solution MUC5AC-E3 is too flexible to be modeled and 3D refinement only show some noisy densities in the expected area.

### MUC5AC-D3 TIL3 structure

The overall structure of MUC5AC-D3 (Fig. 2A) is similar to MUC2-D3 and VWF-D3. However, an overlay of MUC5AC and MUC2 VWD3 assemblies show differences including the loops β4-β5 and β8-β9 in VWD3 domain and the loop β1-β2 in the TIL3 domain, all located on the same side of the molecule (Fig. 2B). The TIL3 domain is the region of the MUC5AC-D3 molecule presenting more differences when compared with MUC2-D3 and VWF-D3 assemblies (Fig. 2C). The MUC5AC-TIL3 β1-β2 loop, unlike the other mucins and VWF, is remarkably rich in arginines (20%) (Fig. 2D). The disulfide bond organization in this domain is unique but still shows characteristics in common with the structures previously described (Dong et al, 2019; Javitt et al, 2020). The disulfide bonds between MUC5AC-TIL3 cysteines 1165–1206, 1189–1228, and 1210–1224 are conserved at similar positions in all three proteins, in addition to the Cys1174 forming the intermolecular dimer disulfide bond. The Cys1181 interacts with the C8-3 region in MUC5AC and MUC2 (Cys1137) stabilizing the domain organization while the equivalent cysteine in VWF (Cys1149) is involved in an extra interdomain disulfide bond. On the other hand, MUC5AC and VWF have an identical disulfide bond in the center of the loop β1-β2 that is absent in MUC2, Cys1185-Cys1196 in MUC5AC and Cys1153-Cys1165 in VWF. MUC2 contains an N-glycosylation site in this area, Asn1154, not present in the other mucins or VWF. The MUC5B-TIL3 domain shares 74.2% identity with MUC5AC and almost certainly displays the same disulfide bond arrangement.

There are also remarkable differences in the relative position of the whole TIL3 domain in relation to VWD3. In MUC2 and VWF, the center of the TIL3 β1-β2 loop is located only 6.2 Å and 8.8 Å away from VWD3 β9, while in MUC5AC they are separated by 14.4 Å. This close contact is explained by the presence of multiple hydrogen bonds between these domains in MUC2 (Gln956-Glu1161, Glu961-Ser1153 and Lys979-Tyr1159) and VWF (Glu954-His1159, Ser958-Ala1152, and Ser958-His1174) that are completely absent in MUC5AC.

### MUC5AC-D3 open conformation

During the 3D-classification processes, a second MUC5AC-D3 assembly conformation was observed. These were estimated to

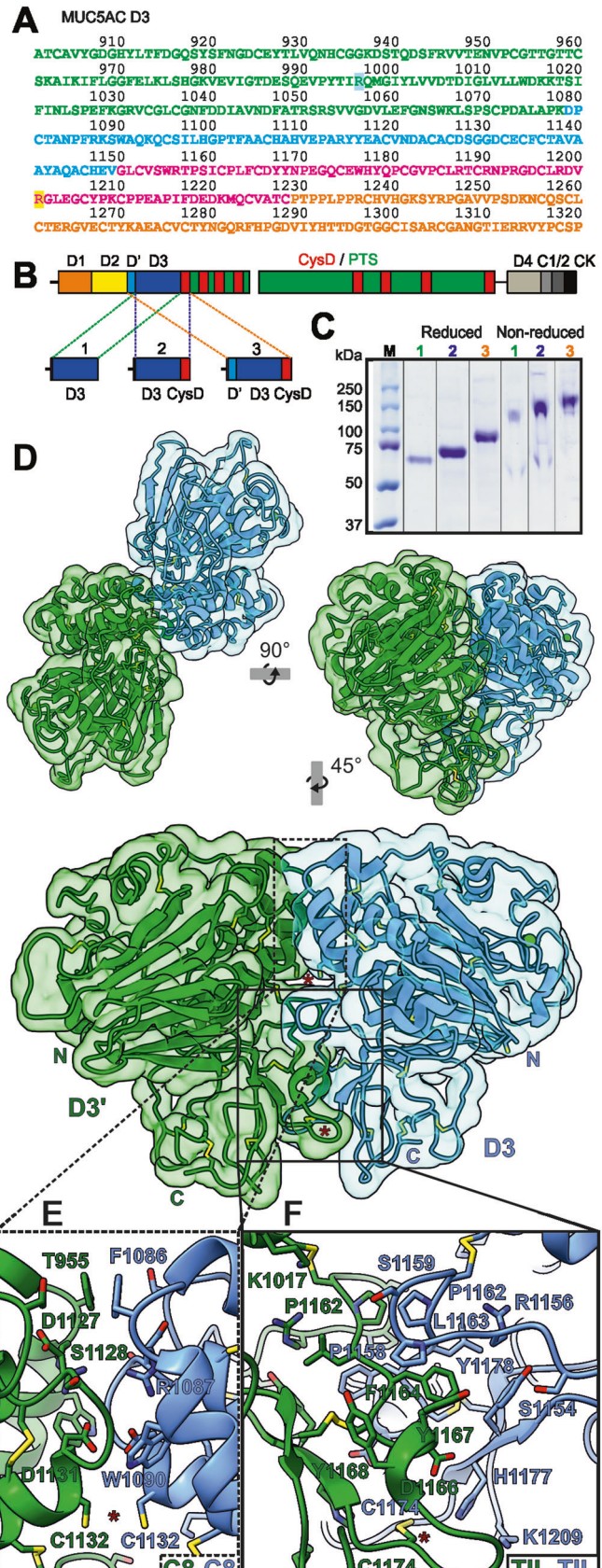

**A**  MUC5AC D3

(sequence panel A)

**B**  D1 D2 D' D3 — CysD / PTS — D4 C1/2 CK

**C**  Reduced  Non-reduced

**D**

**E**  C8-C8

**F**  TIL-TIL

**Figure 1. CryoEM structure of the MUC5AC-D3 assembly.**

(A) MUC5AC-D3 sequence. VWD3 sequence is showed in green, C8-3 in blue, TIL3 in pink and E3 in orange. The residues affected by the SNPs rs36189285 (R996) and rs878913005 (R1201) are highlighted in blue and yellow respectively. (B) Schematic sketch of the domains of MUC5AC mucin with the N-terminal region (VWD1 (orange), VWD2 (yellow), VWD' (light blue) and VWD3 (dark blue)), nine CysD domains (red) surrounded by PTS sequences densely *O*-glycosylated to form mucin domains (green) and the C-terminal region (VWD4 (light gray), VWCs (dark gray) and CK (black)). The fragments analyzed are marked 1, 2, 3. (C) SDS-PAGE analysis of reduced and non-reduced D3 (1), D3-CysD (2) and D'-D3-CysD (3) reveals the formation of reducible dimers in all three fragments. (D) MUC5AC D3 assembly cryoEM density map and cartoon representation showing the disulfide bonds. The map and model of the two monomers are shown in green and cyan. The $Ca^{2+}$ ions are shown as green spheres. The top left figure represents the top view of the molecule. It is rotated anticlockwise by 90˚around the x-axis in the top right figure, and rotated clockwise by 45˚around the y-axis and enlarged by 50% in the bottom figure showing the details of the front view. Putative intermolecular disulfide bonds are marked by red stars (Cys1132-Cys1132' bond seems to be reduced). N-terminal (N) and C-terminal (C) of each monomer are marked. (E) Detail of the MUC5AC-D3 covalent dimerization interface zoomed in from (D) showing the interaction between C8-3 domains. (F) Detail of the MUC5AC-D3 covalent dimerization interface zoomed in from (D) showing the TIL3-TIL3' interaction. Source data are available online for this figure.

represent 17% of the total MUC5AC particles. The 3D volume refined with these particles showed low resolution and quality due to a severe preferred orientation problem. The particles were 2D classified showing all classes having the same top view orientation (Fig. 3A). The maximum diameter of the new open conformation classes is 30% higher than in the standard or closed conformation. The 2D top view of both conformations display enough details to recognize the VWD3 domain β-sandwich and the C8-3 domain, and together with the low-resolution 3D map allowed us to generate a tentative model of this conformation (Fig. 3B and Appendix Fig. S2). The VWD-C8 interaction dissociates and exposes a hydrophobic surface in VWD3 to the solvent (Fig. 3C). The domains establish a novel interaction between the loops β7-β8 and β11-β12 in VWD3 and the C-terminal side of α4 in C8-3, and between the VWD3 β8 and β1-β2 TIL3 loop (Fig. 3D). The long connecting loop between VWD3 and C8-3 domains makes this conformational change possible (Fig. 3B). This conformational change could explain the previous observation that MUC5AC binds significantly more to hydrophobic surfaces compared to MUC5B (Carpenter et al, 2021).

Recently an open structure from the D10 assembly in FCGBP has been published (Yeshaya et al, 2024), supporting the idea that the opening of the VWD assemblies could have a physiological function. However, there are multiple differences between these two structures (Fig. 3E). The FCGBP structure was solved by crystallography and therefore the crystal packaging could promote or alter the structure of the open conformation. The relative position of the domains is completely different. In the MUC5AC-D3 closed conformation the N-terminal of VWD and the C-terminal of C8 points in the same direction and are only 12 Å apart. In FCGBP they are 55 Å apart and rotated 90°, while in MUC5AC open conformation they are at 34 Å and rotated −45°. Furthermore, in FCGBP the domains interact only through the connecting loop as there are no other intramolecular interactions in

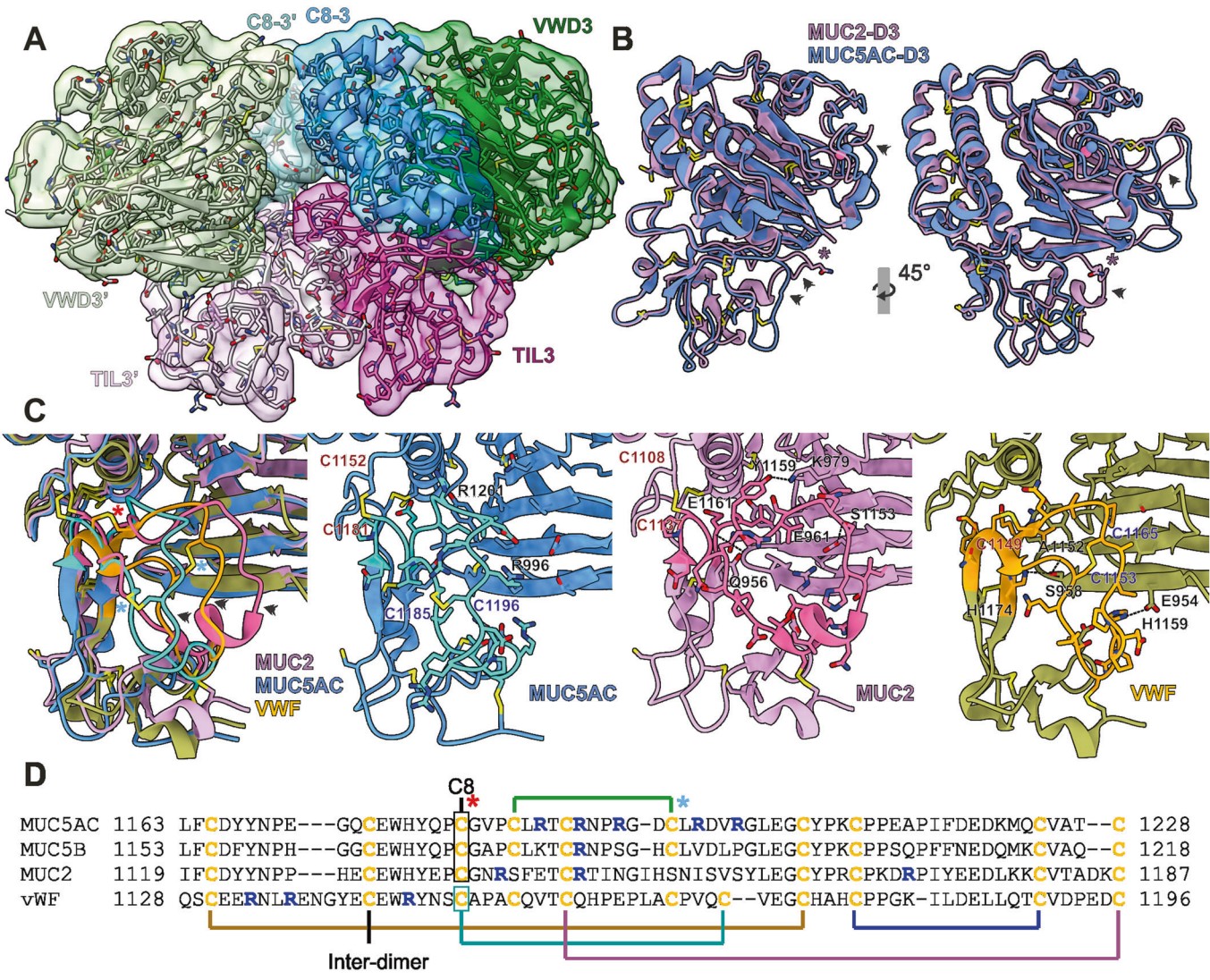

**Figure 2. MUC5AC-D3 domains.**

(A) Front view of MUC5AC-D3 assembly dimer from cryoEM density map and model. The two VWD3 domains are shown in dark and light green, C8-3 domains in dark and light blue, and TIL3 domains in dark and light pink. (B) Structural alignment of MUC5AC-D3 (blue) and MUC2-D3 (pink. PDB code: 6rbf). Left image is presented in the same orientation as in (A). Right image is rotated clockwise 45°. Regions with high variability are marked by black arrows. The *N*-glycosylated Asn1154 in lateral chain of MUC2 is marked with a black star. (C) Detail of TIL3 structurally aligned of MUC5AC-D3 (blue), MUC2-D3 (pink. PDB code: 6rbf), and VWF-D3 (orange. PDB code: 6n29). The TIL3 β1-β2 loop is highlighted in brighter colors. To the left, superposition of all three structures showing the larger distance between the TIL3 and VWD3 domains in MUC5AC. The distinct disulfide bonds are marked by stars, in red the one connecting the TIL3 domain with C8-3 domain and in blue the internal TIL3 β1-β2 loop disulfide bond. All three structures are shown separately showing the TIL3 β1-β2 loop and interfacing residues lateral chains. The cysteines involved in the distinctive disulfide pattern are labeled. Hydrogen bonds between VWD3 and TIL3 are showed with dashed black lines and the residues involved are annotated. In MUC5AC, the residues affected by SNP variation at the amino acids R996 and R1201 are marked. Regions with high variability are marked by black arrows. (D) Amino acid sequence alignment of MUC5AC, MUC5B, MUC2, and VWF TIL3 domains. Disulfide bonds are marked. Stars mark distinct disulfide bonds as in (C). Cysteines are colored yellow and arginines in blue.

contrast to MUC5AC (Fig. 3D). The FCGBP-D10 structure lacks the TIL domain known to interact with the VWD domain in other VWD assemblies to maintain the compact structure where the conformational changes were explained by the GDPH autocatalytic cleavage not present in MUC5AC-D3 (Yeshaya et al, 2024).

## MUC5AC-D3 variants

Interestingly, when genomic databases of MUC5AC were analyzed, the parts of MUC5AC most different from the other mucins as

discussed above (Fig. 2) contained two variants where a single amino acid had been replaced (Fig. 4A). In both of these, one of the arginines typical for MUC5AC was replaced by a glutamine (Arg996Gln, rs36189285) or by a tryptophan (Arg1201Trp, rs878913005). We produced recombinant proteins with each variant separate as well as with both arginines replaced. The structure of each of these were solved by cryoEM and compared (Appendix Table S1; Appendix Figs. S3 to S5). The general appearance of the dimers is essentially identical (Fig. 4B). However, careful comparison of each variant to the WT elucidates the effects

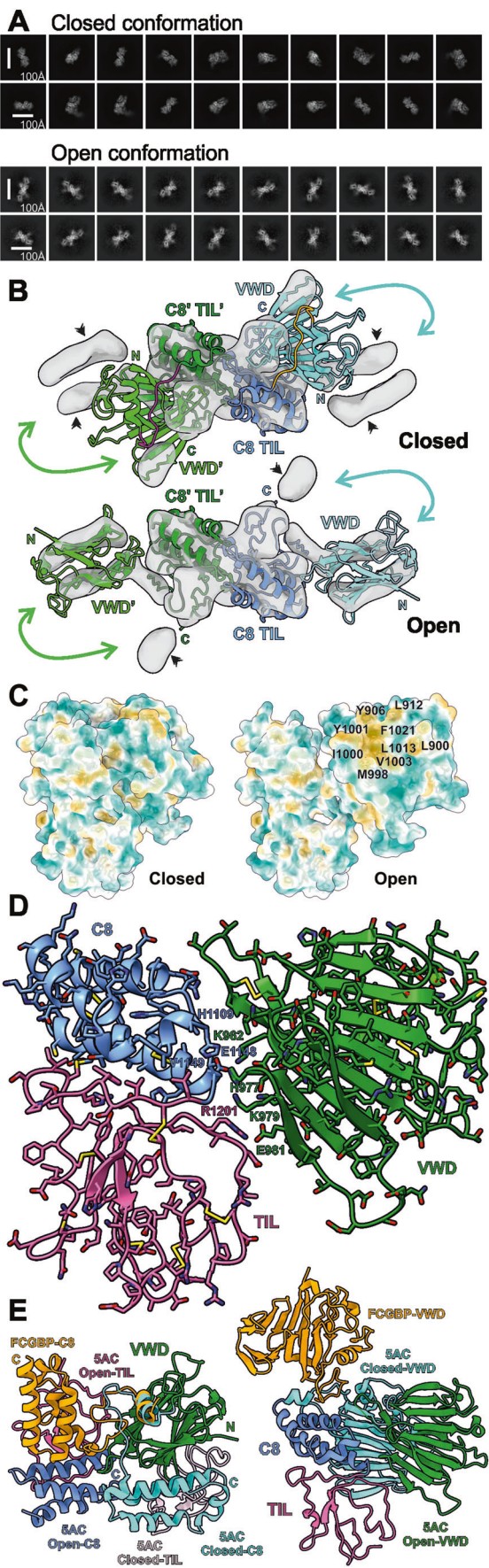

**Figure 3. MUC5AC-D3 assembly open conformation.**

(**A**) CryoEM 2D classes, box size 220 Å. The top figure shows the closed conformation 2D classes from the high-resolution structure shown in Figs. 1 and 2. The discarded particles from an initial 3D classification were further 2D classified. These 2D classes are shown in the bottom panel, open conformation. (**B**) CryoEM low-resolution map generated using the particles from (**A**) bottom panel. The top figure shows the fitting on the closed conformation model and the bottom the proposed model for the open conformation. The VWD3 domain of one monomer is shown in light green, C8-3 and TIL3 domains in dark blue and the connecting loop in magenta. The VWD3 domain of the other monomer is shown in cyan, C8-3 and TIL3 in blue and the connecting loop in orange. The black arrows show the densities not covered by the models. The non-occupied densities in the closed form are explained by the movement of VWD3 as shown by the cyan arrows. The C-terminal of TIL3 points toward the marked densities in the open conformation as they could represent E3 and/or CysD. (**C**) Surface representation of the closed (left) and open (right) conformation colored by molecular lipophilicity potential (MPL) from dark cyan (most hydrophilic) via white to dark goldenrod (most lipophilic). The newly exposed hydrophobic pocket residues in the open conformation are labeled. (**D**) Detail of proposed model for MUC5AC-D3 open conformation. VWD3 is colored in green, C8-3 in blue and TIL3 in pink. Amino acids K962 and E1148, and E981 and R1201 that could form putative salt bridges, respectively, are marked, as are K979 and V1149 that could form a hydrogen bond. These residues and the interfacing histidines His977 and His1109 are labeled. (**E**) MUC5AC-D3 closed and open conformation and FCGBP D10 alignment (Yeshaya et al, 2024). In the left figure MUC5AC-D3 closed (C8-3 in cyan and TIL3 in light pink) and open (C8-3 in blue and TIL3 in dark pink) conformation and FCGBP D10 (C8-10 in orange) were aligned to the VWD domain (green). The N-terminal (N) of VWD and C-terminal (C) of the different C8 domains are marked following the same color code. In the right figure MUC5AC-D3 closed (cyan) and open (green) conformation and FCGBP D10 (orange) were aligned by the C8 (blue) and TIL (pink) domains.

of the mutations (Fig. 4A). The Arg996Gln mutation, located in VWD3 β9, is placed directly in the interface between the VWD3 and the TIL3 domain. The substitution results in a slight reduction of the distance between the two domains (Fig. 4A). No interactions involving this residue are observed in the WT or in Arg996Gln, but considering the resolution limitations a hydrogen bond between Arg996 or Gln996 and a TIL3 β1-β2 loop residue cannot be completely ruled out. The observed alteration can also be attributed to arginine steric repulsion.

In contrast, the Arg1201Trp mutation, placed in the β1-β2 TIL3 loop, only affects the adjacent residues. The solvent exposed arginine is substituted by a tryptophan pointing toward the VWD-C8/TIL interface. The tryptophan accommodates into the VWD3 hydrophobic interfacing region in close proximity to Leu1013. In the double mutant, the effect of the Arg996Gln mutation is more pronounced, bringing the VWD3 and the TIL3 domain closest together.

Even though no major rearrangements were found, both SNPs can affect the equilibrium between the dimer open and closed conformation. Open conformation dimers were constantly found during cryoEM 2D and 3D classifications in the WT assembly. Only when 2D templates from WT open conformation were used for particle picking, these could be detected in the Arg996Gln sample. The same standardized method used in the WT dataset was used to calculate the ratio between the closed and open conformation in all the variants. The ratio in Arg996Gln dataset, 98:2, illustrates the important reduction in open conformation particles from 17% in the WT to only 2%. The Arg1201Trp mutation implied a more discrete reduction in open conformation

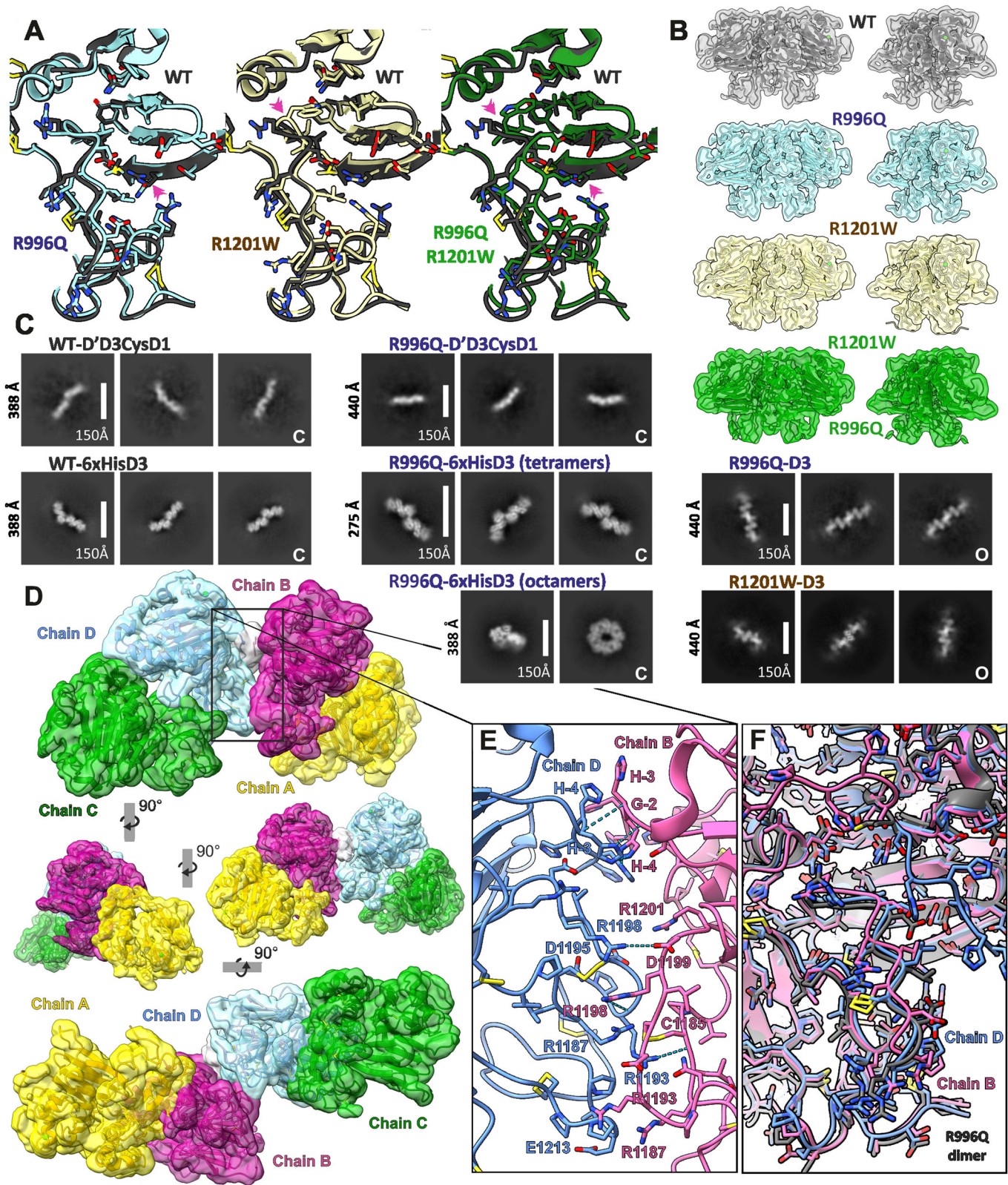

particles showing a ratio of 89:11 between the closed and open conformation. However, it represents a reduction of 35% respect to the WT. The double mutant Arg996Gln Arg1201Trp showed a ratio of 96:4 confirming the effect of the mutations.

## MUC5AC-D3 tetramers

The micrographs of the MUC5AC-D'D3-CysD1 assembly showed covalent dimers (Fig. 2), but also numerous tetrameric particles

◄

**Figure 4. MUC5AC-D3 tetramerization.**

(A) Detail of TIL3-VWD3 interface mutants aligned with the MUC5AC-D3 WT (dark gray). The left figure shows the Arg996Gln mutant in cyan, the mid shows the Arg1201Trp mutant in yellow, and the right shows the double mutant Arg996Gln-Arg1201Trp in green. The mutations are pointed by pink arrows. (B) CryoEM density map and cartoon representation of the MUC5AC-D3 dimeric assemblies WT (gray), Arg996Gln (cyan), Arg1201Trp (yellow), and Arg996Gln-Arg1201Trp (green) at two 45° angles. (C) CryoEM 2D classes of higher order oligomers. Box sizes are specified for every group of classes. The groups of closed conformation oligomers are marked with "C" and the open conformation with "O". (D) MUC5AC-D3 Arg996Gln tetrameric assembly cryoEM density map and cartoon representation. One covalent dimer is shown in yellow (chain A) and magenta (chain B) and the other in green (chain C) and cyan (chain D). The His-tag density is shown in white. The top image shows a lateral view of the tetramer. It is rotated clockwise by 90° around y-axis and reduced 1.5 times in the middle-left figure, rotated clockwise by 90° around y-axis again in the middle-right figure, and rotated anticlockwise by 90° around x-axis and rescaled to the original size in the bottom figure. (E) Detail of the MUC5AC-D3 non-covalent tetramerization interface zoomed from (D). The predicted salt bridges and hydrogen bonds are shown as dashed cyan lines. (F) Structural alignment of the tetramer chain B (pink) and chain D (blue) against the R996Q dimer chain A (gray).

(Fig. 4C). In the same way, all MUC5AC-D3 variants showed some tetramers in cryoEM. The interaction is shown to be flexible as most of the 2D classes are noisy and 3D reconstructions only show elongated blobs, but the presence of the His-tag stabilizes it. The WT 6xHis-MUC5AC-D3 assembly shows 2D classes with structural features just in one orientation and the 3D reconstructions thus have low quality. Similarly, the structure of the tetrameric 6xHis-MUC5AC-D3 Arg996Gln assembly was solved at low resolution (6–7 Å), not enough to reveal the molecular mechanism of the non-covalent oligomerization. Surprisingly, the 6xHis-MUC5AC-D3 Arg996Gln adopts another conformation not observed in the other variants as the protein forms ring shape octamers (Fig. 4C). The ring formation gives stability to the interactions. However, after refinement only one of the tetramers was well defined, the others were poorly defined due to remaining flexibilities and difficulties in isolating octamers from tetramers in some orientations. Particle subtraction followed by local refinement let us solve the structure of a tetramer forming part of the octamer at 3.69 Å resolution (Fig. 4D,E; Appendix Fig. S6, Appendix Table S1). The general appearance and local resolution of the different MUC5AC-D3 dimers and tetramers are shown in Appendix Fig. S7.

The MUC5AC-D3 non-covalent tetramer interaction is driven by the TIL3 domain and involves an interface area of 385.4 Å$^2$ (Fig. 4D). The interaction between the MUC5AC-TIL3 domains occurs in the arginine-rich loop β1-β2 stabilized by a disulfide bond between Cys1185-Cys1196 (Fig. 2C,D). The interaction is mainly hydrophilic (Fig. 4E). Intermolecular salt bridges and hydrogen bonds are formed between lateral chains of Asp1199-chainB (Asp1199B) and Arg1198-chainD (Arg1198D), and between Arg1198B lateral chain and the main chain of Arg1193D and Asp1195D, Arg1193D and Cys1185B, and between Arg1198D and Leu1197B and Arg1198B. The model also shows the presence of the lateral chain of Arg1187D and Arg1193D in the proximity of the Cys1185B-Cys1196B disulfide bond and Arg1198B pointing to the same disulfide bond in chain D.

The comparison between the tetrameric MUC5AC-D3 Arg996Gln assembly and the free closed conformation dimer shows significant differences only in the TIL3 domain region (Fig. 4F). While the tetramer chain D shows only minor deviations, the TIL3 loop β1-β2 in chain B changes its relative position with C8-3 and VWD3 domains allowing the interaction with chain D.

The solved structure confirms the role of the His-tag in the interaction. No interactions between the His-tag of one monomer and the core of the other one were observed, just His-tag–His-tag

interactions, supporting that it is an unspecific interaction that stabilizes the TIL3-TIL3 interface (Fig. 4E).

Moreover, 2D classifications suggest that the MUC5AC-D3 assembly can also form an additional kind of tetramers involving the open conformation (Fig. 4C). This interaction is linear and leads to the formation of high-order oligomers that look longest in Arg996Gln. However, the open conformation oligomers were only found in assemblies lacking D' or N-terminal 6xHis-tag. The 2D classes together with the open conformation low-resolution map (Fig. 3B) suggested a major steric impediment at the N-terminal region. The symmetric interaction of two VWD3 domains through the external side of the β-sheet 1 seems to bury both N-terminal residues. Assemblies containing D' or N-terminal His-tag were only found to form tetramers based in closed conformation covalent dimers, making it likely that the open form tetramers are an artefact.

## MUC5AC-D3 tetramerization and physiological properties

The tetrameric MUC5AC will cross-link linear polymers as illustrated in Fig. 5A. The O-glycosylated PTS domains are extended rods and thus expected to drive the MUC5AC C-termini apart. When the covalent MUC5AC filaments interact, they will interact generating 'net-like' structures (Fig. 5B). However, as the tetramer interaction is non-covalent, one can expect many irregularities, but likely relatively flat sheets are expected. This is in line with what we know of MUC5AC organization from staining of tissues where the MUC5AC mucin is found. The surface mucus of the stomach is largely made up of MUC5AC and when tissue sections stained for MUC5AC a stratified and laminated organization appears (Fig. 5C). Such laminated organization is easiest obtained by sheets. The laminated mucus organization is similar to the one found for the inner mucus layer of the colon made up by the MUC2 mucin (Ambort et al, 2012; Johansson et al, 2008).

Electron microscopy of the tracheal surface illustrates the different organization of the two lung mucins, the linear bundled strands of MUC5B and the more net-like appearances of the MUC5AC (Fig. 5D). In this case the MUC5AC has not been expanded to form sheets. That the two different mucins organize into these two forms is also shown by Carpenter et al when pure MUC5B and MUC5AC mucins were studied by EM (Carpenter et al, 2021).

The VWD3 assembly-mediated covalent dimerization in the MUC5AC mucin described is as predicted by homology with MUC2 and VWF. Together with the C-terminal dimerization

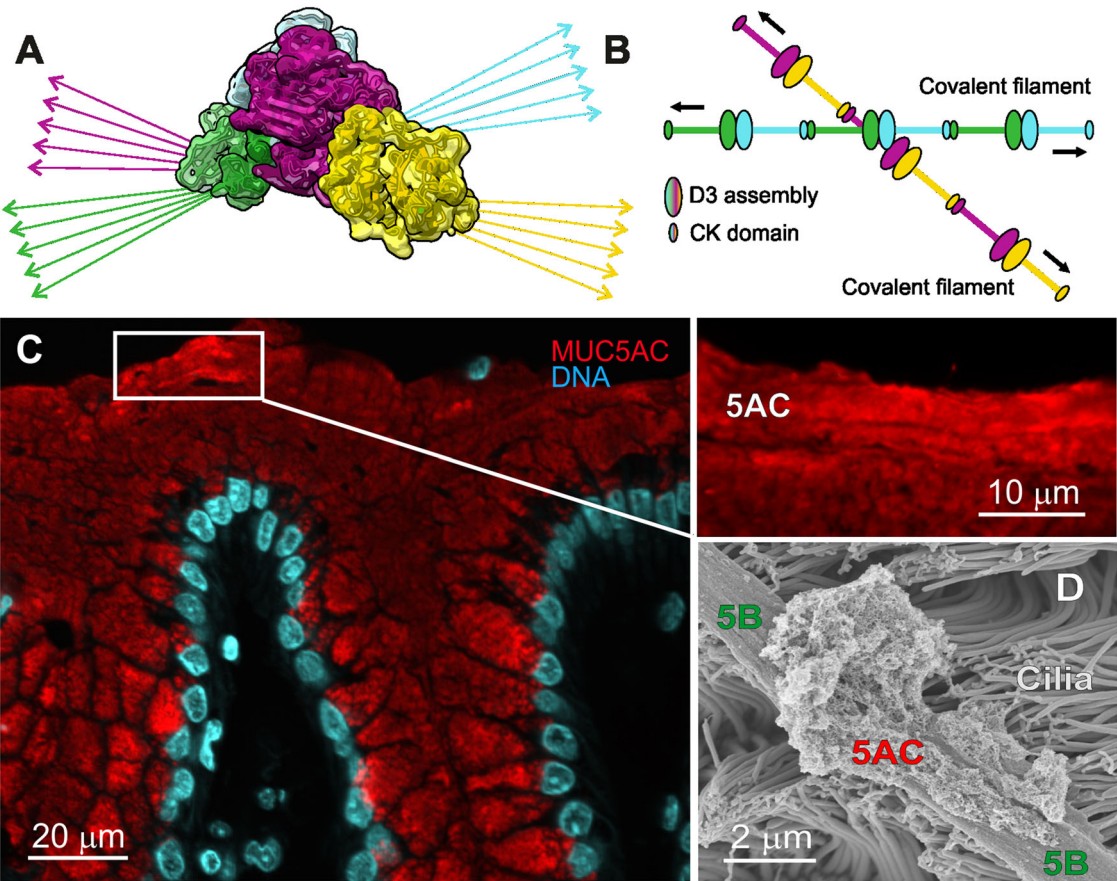

**Figure 5. MUC5AC-D3 physiological properties.**

(A) MUC5AC-D3 Arg996Gln tetrameric assembly cryoEM density map and arrows representing potential directions for the C-terminal PTS domains. One covalent dimer is shown in yellow (chain A) and magenta (chain B) and the other in green (chain C) and cyan (chain D). The arrows colored as their D3 domains are protruding from the TIL3 C-termini. (B) Two schematic MUC5AC covalent linear filaments (C-terminal cystine-knot and D3 covalent dimer) and one non-covalent D3 interaction form a tetramer in the middle. From this model, the formation of repetitive similar interactions will generate the predicted net-like structures. (C) Carnoy fixed human stomach biopsy paraffin section stained with a monoclonal anti-human MUC5AC antibody (45M1; red) and Hoechst (nuclei; blue). The enlarged white square shows stratified surface mucus positive for MUC5AC. One tissue out of 10. (D) Scanning electron micrograph of a piglet airway showing a MUC5B bundled strand, MUC5AC mucus attached to the bundle, and cilia. One tissue out of 20. Source data are available online for this figure.

observed in all gel-forming mucins and VWF, lead to the formation of long linear polymers (Gallego et al, 2023; Javitt et al, 2020). First, the C-terminal inter-disulfide bond is formed in the ER (pH 7.4) and in the trans-Golgi network (pH 6.2) the N-termini are coupled together (Asker et al, 1998; Axelsson et al, 1998). To prevent the N-terminal disulfide bond formation in the ER, the interaction of the covalent oligomerization interface should only happen at lower pH. The dominance of charged residues close to the interface could have a role regulating the formation of the covalent link along the secretory pathway, especially His1177 and Asp1166, conserved in all gel-forming mucins. Different structures of the MUC2-D3 assembly have been solved by crystallography and cryoEM, but all of them at low pH as found in the storage granule (Javitt et al, 2019; Javitt et al, 2020).

To understand the organization of mucins in the secreted mucus, interactions have to be studied at neutral pH. Here the first mucin D3 assembly solved at neutral pH is presented. Interestingly no major structural differences were observed compared to the low pH. It indicates that once the covalent N-terminal dimerization is

stabilized the interface is locked, even if the Cys1132-Cys1132 dimer bond is partially reduced. This disulfide bond reduction was first observed for the equivalent Cys observed in the MUC2 crystallographic structure (Javitt et al, 2020). This was then attributed to radiation damage, an event that could also occur in cryoEM and can provide information about "weak links" that can be of structural significance (Weik et al, 2000). This suggests that the formation of the Cys1132-Cys1132 disulfide bond could be tightly regulated by its redox potential, requiring the higher oxidizing environment found in the Golgi compared to the ER (Kellokumpu, 2019). Its higher radiation damage susceptibility could also be explained by the formation of a stabilizing S···O interaction (Bhattacharyya et al, 2020) with the carbonyl $O$-atom from the same cysteine located just at 3.3 Å from the $S$-atom. The calcium binding site is unaltered at neutral pH and the density maps show signals in the area compatible with the presence of a metal ion despite the present resolution limitations.

Observations of MUC5AC under physiological conditions has shown that it is different from MUC5B in that it does not appear as

long linear polymers (Carpenter et al, 2021; Ermund et al, 2017; Ridley et al, 2016). Instead, it forms more complex and likely net-like structures as shown here and previously (Carpenter et al, 2021; Ryan et al, 2015), requiring the existence of other intermolecular interactions. The here observed capability of MUC5AC to form non-covalent dimers through the TIL3 domain and thus MUC5AC-D3 tetramers explains the molecular mechanism for net-like polymer formation. The interactions are mainly hydrophilic as it is based on multiple hydrogen bonds and salt bridges through a densely positively charged region. Unlike the covalent MUC5AC-D3 dimerization that locks the assembly, the non-covalent formation of tetramers is flexible and can be regulated by pH and ionic strength upon and after secretion. The unique MUC5AC arginine-rich TIL3 domain suggests that this interaction only occurs in MUC5AC.

MUC5AC can appear both in a closed and an open conformation. Although the open model is derived from low-resolution cryoEM, it shows that the VWD assemblies can open in solution supporting the physiological relevance of the previously reported FCGBP VWD10 structure (Yeshaya et al, 2024). Considering other known VWD3 assemblies, we cannot exclude the existence of VWF or MUC2-D3 open conformations, but the closer interactions between the TIL domain and VWD make them less energetically favorable. The open conformation implies the exposure of a highly hydrophobic surface that is not evident in FCGBP-D10. The open conformation tetramers and high-order oligomers found in MUC5AC-D3 suggest that the external site of the VWD3 β-sheet 1 becomes a new interaction surface that could establish contacts with other mucins, mucus-associated proteins, or other domains from the same molecule. This could explain the hydrophobic properties attributed to the mucins but does not explain any of the actual structures.

In addition to the observed non-covalent interactions within the VWD3 assembly explaining the net-like appearance of MUC5AC, there are additional possible interactions within mucins. The CysD2 domain of MUC2 was recently shown to form weak homotypic dimers that was further stabilized by transglutamination catalyzed by TGM3 (Recktenwald et al, 2024). Human MUC5AC contains nine CysD domains, thus similar mechanisms could take place in MUC5AC to produce a highly cross-linked mucin. Although all the CysD domains have a common structure, there is a large variation in especially surface amino acids opening for variation in the interactions.

SNPs affecting the respiratory gel-forming mucins have been previously associated with disease (Sabo et al, 2023; Seibold et al, 2011; Shrine et al, 2019). However, mucin sequencing has proved to be challenging due to their highly repetitive sequences in the PTS domain (Svensson et al, 2018). Mucin sequences and SNPs have been poorly annotated in these and flanking regions, including the VWD3 assembly. Here we observed that two frequent coding missense SNPs in the MUC5AC-D3 assembly, rs36189285 and rs878913005, are located in the interaction surfaces involved in the MUC5AC tetramerization. The rs36189285 SNP corresponds to Arg996Gln, while rs878913005 corresponds to Arg1201Trp, both substituting the for MUC5AC typical arginines. Arg996Gln is present in 32.9% of East Asian population and in 0.3% of European population in the Genome Aggregation Database (gnomAD), whereas Arg1201Trp is present in 18.6% of the European population and only 0.6% in the Asian population. A high

prevalence of double mutants is not observed except in Finnish population, where both mutants are present independently in 10.9% and 23.9% (Karczewski et al, 2020).

The two SNPs (Arg996Gln and Arg1201Trp) can be involved in the closed-open conformation equilibrium and the described non-covalent oligomerization. However, no wide-ranging conformation differences were observed in the dimeric VWD3 mutants (Fig. 4B), although minor alterations could be overlooked due to the low resolution in the flexible open conformation. Anyhow, the two mutations are likely to affect the molecular dynamics rather than the structure.

The mutation Arg996Gln, frequent in Asian population, affects the distance and the dynamics between the TIL3 domain and the VWD3, directly interfering in the conformational change and oligomerization. The open conformation was not detected in the D'D3CysD1 Arg99Gln cryoEM preparation during standard particle picking. Only template picking based on the WT open structure revealed a few particles. This observation, together with the described reduction of the distance between VWD3 and TIL3 domains, suggests that this mutation stabilizes the closed conformation and reduces the steric repulsion between Arg996 and TIL3. However, the VWD3 Arg996Gln mutant showed a higher occurrence of high-order oligomers with an open conformation (Fig. 4C), supporting that the open conformation can still occur. When it forms, the absence of the positive charge residue near the newly exposed hydrophobic interaction face can modulate its properties. On the other hand, the Arg996Gln mutation produces a dramatic conformational change in the standard closed conformation tetramer. The His-tagged Arg996Gln mutant showed a high proportion of octameric particles in cryoEM that were essential for solving the high-resolution structure of the non-covalent interaction. The TIL3-TIL3 interaction is flexible in all cases, but the Arg996Gln mutation could add more freedom to allow the octameric arrangement. This mutation probably also increases the interaction affinity and therefore stabilizes the TIL interface, consequently also the second TIL3 domain in each disulfide-bonded dimer (Fig. 4D) appears involved in non-covalent interactions making the formation of octamers possible.

The Arg1201Trp SNP, frequent in European population, shows a double role in the conformation alteration of the VWD3 assembly. In the closed conformation it establishes a novel interaction between the TIL3 domain and VWD3, and the open conformation removes the possibility of a putative salt bridge between Arg1201 and Glu981. By that we can speculate that this mutation should favor the closed conformation, even though open conformation molecules were observed in cryoEM. The role of Arg1201Trp in MUC5AC-D3 tetramerization is not fully understood, but Arg1201 is located at the interaction site and not directly involved in any contact. However, its close proximity to Asp1195, limitations in resolution, and flexibility of the interaction make it impossible to disregard the importance of the Arg1201 amino acid in the formation of the interface. On the other hand, the presence of the tryptophan interacting with VWD3 could affect the TIL3 flexibility important for the interaction. The double mutant shows that once Arg996 is mutated, Trp1201 stabilizes the interface bringing TIL3 closer to VWD3. In theory, Arg1201Trp should limit the TIL3 flexibility and have the opposite effect of Arg996Gln. Nevertheless, we found that the Arg1201Trp SNP could favor tetramerization and thus net-like formation. This mutation has also

recently been found to be associated with dysregulated inflammatory responses across keratoconic cone where MUC5AC mucin plays a protective role (Jaskiewicz et al, 2023). Arg996Gln and Arg1201Trp have thus the same effects in the closed-open conformation equilibrium, stabilizing the closed form. In the same way, the effect of both mutations in tetramerization seems to be equivalent even if the outcome has not been completely elucidated.

That the Arg1201Trp SNP promotes tetramerization suggests that it should increase mucus crosslinking. The amount of MUC5AC is known to be increased at lung diseases and especially COPD (Fahy et al, 2010; Fernandez-Blanco et al, 2018; Radicioni et al, 2021) and its capacity trapping bacteria could be increased (Ermund et al, 2021). MUC5AC is known to anchor the MUC5B bundled strands to the surface goblet cells and a more cross-linked MUC5AC might increase the attachment and retard mucus clearance (Ermund et al, 2017). The structure of the MUC5AC VWD3 domain is thus implied as important for understanding mucus properties, especially at disease. The presence of amino acid variations at the interaction surfaces for MUC5AC tetramerization suggests that these could affect susceptibility and severity of diseases related to mucus.

# Methods

### Reagents and tools table

| Reagent/Resource | Reference or Source | Identifier or Catalog Number |
|---|---|---|
| **Recombinant DNA** | | |
| pCEP-His | ThermoFisher Scientific | Cat #V04450 |
| **Antibodies** | | |
| Monoclonal mouse anti-MUC5AC | Abcam | Cat# ab3649 |
| Polyclonal donkey anti-rabbit Alexa Fluor 488 | ThermoFisher Scientific | Cat# A-21206 |
| Polyclonal donkey anti-mouse Alexa Fluor 647 | ThermoFisher Scientific | Cat# A-31571 |
| **Chemicals, Enzymes and other reagents** | | |
| Freestyle™ CHO | Gibco | Cat# 12651014 |
| NovaCHOice transfection kit | Merk | Cat# 72622 |
| Mini-PROTEAN® TGX™ precast protein gels | Bio-Rad | |
| Hechts 34580 | ThermoFisher Scientific | |
| Estrumate | Intervet GmbH | |
| Ursotamin | Serumwerk Bernburg | |
| Stresnil | Elanco Animal Health | |
| Tanax T61 | Intervet GmbH | |
| Perfadex | XVIVO Perfusion | |
| Conductive silver paint | Ted Pella | Cat# 16040-30 |
| **Software** | | |
| EPU software | ThermoFisher Scientific | |
| cryoSPARC v.4.2.1 and 4.6.0 | https://cryosparc.com | |

| Reagent/Resource | Reference or Source | Identifier or Catalog Number |
|---|---|---|
| Topaz v.0.2.3 | https://cb.csail.mit.edu/topaz | |
| AlphaFold2 | https://alphafold.ebi.ac.uk/ | |
| Molrep v.11.0 | https://www.ccp4.ac.uk/html/molrep.html | |
| Coot v. 0.9.8.1 | https://www2.mrc-lmb.cam.ac.uk/personal/pemsley/coot | |
| Phenix v.1.20.1-4487 | https://phenix-online.org | |
| Molprobity v.4.5.2 | http://molprobity.biochem.duke.edu | |
| Pymol v.4.6.0 | https://www.pymol.org | |
| ChimeraX v.1.6.1 | https://www.cgl.ucsf.edu/chimerax | |
| ZEISS ZEN Microscopy Software | Carl Zeiss | RRID:SCR_013672 |
| Imaris v.9.5 | Oxford Instruments, Abingdon, UK, RRID | RRID:SCR_007370 |
| **Other** | | |
| Durapore Membrane Filter | Millipore | |
| ÄKTA purifier | GE Healthcare | |
| HiTrap chelating HP | Cytiva | Cat# 10431065 |
| HiPrep Q HP 16/10 | GE Healthcare | |
| Mono Q HR 10/10 | GE Healthcare | |
| Superose 6 10/300 | GE Healthcare | |
| Superfrost plus slides | Radnor Township | Cat# 631-9483 |
| ImmEdge® Hydrophobic Barrier PAP Pen | Vector laboratories | |
| UltrAuFoil R1.2/1.3 300# | SPT Labtech | |
| Quantifoil Cu 1.2/1.3 300# | SPT Labtech | |
| GloCube Plus | Quorum | |
| Vitrobot Mark IV | ThermoFisher Scientific | |
| Titan Krios | ThermoFisher Scientific | |
| K2 Summit 4k x 4k detector | Gatan | |
| ZEISS LSM900 Airyscan 2 | Carl Zeiss | |
| Specimen pin stubs | Agar Scientific | Cat# AGG301 |
| Carbon tabs | Agar Scientific | Cat# AGG3347N |
| Zeiss DSM 982 Gemini | Carl Zeiss | |

## Production and purification of MUC5AC plasmids

The recombinant MUC5AC-D'-D3-CysD (WT and Arg996Gln), MUC5AC-D3-CysD (WT) and MUC5AC-D3 (WT and Arg996Gln) (GenBank accession number NM_001304359, residues 800–1481, 901–1481, 901–1366) plasmids were expressed with an N-terminal Hisx6 tag and a C-terminal Myc tag using the mammalian episomal expression vector pCEP-His. MUC5AC-D3

(WT, Arg996Gln, Arg1201Trp, and Arg996Gln Arg1201Trp) was also expressed without any tag using the same expression vector.

CHO-Lec 3.2.8.1-S (Nilsson et al, 2014) cells were grown in 300 ml Freestyle™ CHO with 8 mM L-glutamine in an Erlenmeyer flask in 5% $CO_2$. Transfection with NovaCHOice transfection kit (Merck, Nottingham, UK) was performed according to the manufacturer's instructions. Four hours after transfection the temperature was decreased to 31 °C. The supernatant was harvested after 48 h by centrifugation for 10 min at 200 × g at room temperature and then immediately dialyzed against PBS 10 mM imidazole (His-tagged proteins) or 20 mM Tris pH 8 (His-tag free proteins) at 4 °C. Several MUC5AC-D3 batches of 300 ml each were made in CHO-Lec 3.2.8.1-S.

MUC5AC was filtered (Durapore® Membrane Filter, 0.22 µm GVWP, Millipore) and further purified using an ÄKTA purifier (GE Healthcare). The His-tag-containing proteins were loaded onto a HiTrap chelating HP nickel affinity 1-ml column (Cytiva). The bound components were eluted with a gradient of 10–300 mM imidazole in 20 mM Tris pH 7.4 and 150 mM NaCl. The His-tag free MUC5AC-D3 variants were loaded onto a HiPrep Q HP 16/10 anion exchange chromatography column (GE Healthcare) and eluted in a 0–500 mM NaCl gradient in 20 mM Tris pH 8.

The protein-containing fractions were dialyzed against 20 mM Tris (pH 7.4) and 50 mM NaCl, loaded onto a Mono Q™ HR 10/10 anion exchange column (GE Healthcare) and eluted in a linear gradient from 50 to 500 mM NaCl. It was followed by size exclusion fractionation on a Superose 6 10/300 column (GE Healthcare) eluted in 20 mM Tris pH 7.4, 50 mM NaCl and 10 mM $CaCl_2$ and collected in fractions of 0.5 ml.

Protein purity was checked by SDS-PAGE using 4-15% Mini-PROTEAN® TGX™ precast protein gels (Bio-Rad). Proteins were diluted in 2x SDS-PAGE loading buffer reaching a final concentration of 100 mM DTT or in DTT-free buffer, heated 5 min at 95 °C and loaded into the gel together with the Precision Plus Protein Unstained Standard (Bio-Rad). Gels were stained with Coomassie brilliant blue G-250.

## Single particle cryoEM

In order to optimize the sample homogeneity only the central fraction of the size exclusion was used for all recombinant proteins sample preparation (MUC5AC-D'-D3-CysD1 (WT and Arg996Gln), MUC5AC-6xHis-D3 (WT and Arg996Gln) and MUC5AC-D3 (Arg1201Trp and Arg996Gln-Arg1201Trp). The protein concentration was adjusted to 0.6 µM in 20 mM Tris pH 7.4, 150 mM NaCl, and 10 mM $CaCl_2$.

The samples were loaded onto UltrAuFoil R1.2/1.3 300# (SPT Labtech) holey gold grids (D3 Arg1201Trp, D3 Arg996Gln-Arg1201Trp, 6xHis-D3 Arg996Gln and 6xHis-D3 WT) or Quantifoil Cu 1.2/1.3 300# (SPT Labtech) copper grids (D'-D3-CysD1 WT and D'-D3-CysD1 Arg996Gln) that were previously glow discharged at 15 mA for 40 s with a negative charge in a GloCube Plus (Quorum) glow discharger system. The grids were plunge frozen using a Vitrobot Mark IV (ThermoFisher) set at 100% humidity and 4 °C. 6xHis-D3 Arg996Gln data collections were performed in a Titan Krios microscope (Thermo Fisher) at 0.83 Å/pix and 300 kV acceleration voltage using a K2 Summit 4k x 4k detector (Gatan). 40 frames per movie were collected with an average electron dose per frame of 1.15 e/Å² and nominal defocus between

-0.5 and -3.5 µm in 0.3 µm steps. The data from five grids collected using the same conditions were merged for reconstruction. The datasets from the other five recombinant proteins were collected using EPU software (Thermo Fisher Scientific) in Aberration-free image shift (AFIS) mode at 0.86 Å/pix and 300 kV acceleration voltage and a K3 Summit 6k x 4k detector (Gatan). 40 frames per movie were collected with an average electron dose per frame of 1.25 (6xHis-D3 WT), 1.26 (D3 Arg996Gln Arg1201Trp), 1.29 (D'-D3-CysD1 WT and D'-D3-CysD1 Arg996Gln) or 1.49 e/Å² (D3 Arg1201Trp) and defocus between -0.5 and -2.5 µm (D'-D3-CysD1 WT, 6xHis-D3 WT and D3 Arg996Gln Arg1201Trp) or -0.5 and -3.0 µm (D'-D3-CysD1 Arg996Gln and D3 Arg1201Trp). These parameters are summarized in Appendix Table S1. The micrographs were imported in cryoSPARC v.4.2.1 (Punjani et al, 2017) and patch motion corrected. CTF parameters estimation was performed in cryoSPARC using Patch CTF. The micrographs were manually curated and outliers were removed. The data processing steps leading to the generation of the final models in cryoSPARC v.4.2.1 are schematically represented in Appendix Fig. S1 to S6.

### D'-D3-CysD1 WT

A first round of manual picking was performed. The particles were 2D classified and the best classes were used for template-based automatic particle picking. The particles were extracted using a 256 pixel box size and 2D classified. The junk particles were removed and the rest were used for ab-initio reconstruction (2 classes) and heterogeneous refinement. A fraction of the particles from the best locking class was used for Topaz particle picking training (Bepler et al, 2019). The Topaz picking model was applied to the curated dataset and the particles were extracted using a 256 pixel box size and 2D classified for junk particles removal. The particles were used for ab-initio reconstruction (4 classes) and heterogeneous refinement. A fraction of the particles from the best locking class was used for a new Topaz particle picking training. A new particle extraction was performed and the previous steps were repeated generating new 4 ab-initio classes that were used again in heterogeneous refinement. The heterogeneous refinement classes were used as templates for non-uniform refinement using C1 and C2 symmetries. The particles from the highest resolution class in the non-uniform refinement with C2 symmetry were further 3D classified by a new ab-initio reconstruction round in 2 classes and heterogeneous refinement. The best quality class was non-uniform refined imposing C2 symmetry that was used as an input for a final local refinement also imposing C2 symmetry. The discarded particles from 2D and 3D classification corresponding to the open conformation were curated and used for Topaz particle picking training and extraction. The particles were extracted using a 256 pixel box size and 2D classified for junk particles and closed conformation particles removal. The selected particles were used for ab-initio reconstruction followed my homogeneous refinement and non-uniform refined imposing C2 symmetry.

### D'-D3-CysD1 Arg996Gln

The same protocol as for D'-D3-CysD1 WT Dimers was applied until the first Topaz particle picking and extraction. Then the particles were directly classified in 2 ab-initio classes. The particles from the best class were 2D classified and only the high-resolution classes were selected. These particles were used for a new Topaz training and the process was repeated. The particles were used in a

new ab-initio reconstruction round in 2 classes and heterogeneous refinement. The best quality class was non-uniformly refined imposing C2 symmetry that was used as an input for a global CTF refinement and a final local refinement also imposing C2 symmetry. To look for the presence of open dimers in the sample, D'-D3-CysD1 WT open dimer templates were used for template-based automatic particle picking. Particles were extracted using a 256 pixel box size and 2D classified for junk particle removal. The remaining particles were 3D classified by ab-initio reconstruction in 2 classes and heterogeneous refinement, and further 2D classified.

### D3 Arg1201Trp

An initial blob picking was performed using 1000 micrographs setting a minimum particle diameter of 70 Å and maximum of 140 Å. The particles were 2D classified and the bests classes were used for template-based automatic particle picking. The particles were extracted using a 256 pixel box size and 2D classified. The junk particles were removed and the rest were used for ab-initio reconstruction (2 classes) and heterogeneous refinement. A fraction of the particles from the best locking class was used for Topaz particle picking training. The Topaz picking model was applied to the curated dataset and the particles were extracted using 256 pixels box size and 2D classified for junk particle removal. Particles were divided into two ab-initio classes. A fraction of the particles from the best locking class was used for a new Topaz particle picking training. A new particle extraction was performed and the previous steps were repeated generating new 2 ab-initio classes that were used in heterogeneous refinement. The particles from the highest quality class were 3D classified again by ab-initio reconstruction in two classes and heterogeneous refinement. The best class was non-uniform refined imposing C2 symmetry that was used as an input for a global CTF refinement and a final local refinement also imposing C2 symmetry.

### D3 Arg996Gln Arg1201Trp

The same protocol than in D'-D3-CysD1 WT Dimers was applied changing the ab-initio and heterogeneous refinement with 4 classes for 2 classes and skipping the further repetition.

### 6xHis-D3 Arg996Gln

The same protocol as for D'-D3-CysD1 R996 Dimers, but using a 450 pixel box size, was applied until the ab-initio reconstruction following the first Topaz particle picking and extraction step. Both ab-initio classes, one representing the tetrameric conformation and the other the octameric, were further 2D classified to remove junk particles and used independently in a second Topaz training. Both models were used for particle picking and two group of particles were newly extracted. The particles based in the tetrameric conformation training were 3D ab-initio reconstructed in 2 classes and further heterogeneously refined. The particles from the heterogeneous refinement class corresponding to a tetramer were 3D ab-initio reconstructed in 2 classes and further heterogeneously refined again. The best quality class was non-uniformly refined. Finally, the non-uniform refinement was used as an input for a final local refinement. In parallel, the particles based in the octameric conformation training were also 3D ab-initio reconstructed in 2 classes and further heterogeneously refined. The particles from the heterogeneous refinement class corresponding to an octamer were

3D ab-initio reconstructed in 2 classes and further heterogeneously refined again. The best quality class was non-uniformly refined. The 3D volume was used to create two masks in Chimera (Pettersen et al, 2004), one including the best-defined tetramer in the octameric assembly (A) and the inverted mask (B). The masks were newly imported in cryoSPARC and the mask A was dilated 6 Å and soft padded 8 Å. The particles from the non-uniform refinement were 3D classified in 4 classes using the non-uniformly refinement mask as a solvent mask and the mask A as focused mask. All four classes were further non-uniform refined. The particles and volume from the best resolution class were used for particle subtraction applying mask B. The newly generated particles together with the non-uniform refinement volume and the mask A were used as inputs in a last local refinement step.

### D'-D3-CysD1 WT, D'-D3-CysD1 Arg996Gln, D3 Arg1201Trp, D3 Arg996Gln and 6xHis-D3 WT Tetramers

Manual and template picking was performed as for most of the dimers but using a bigger box size, 320, 450 or 512 pixels. Particles were 2D classified and the best classes were used for Topaz particle picking. Particles were extracted using the same box size and further 2D classified.

## Closed-open conformation ratio

In order to determine the closed-open conformation ratio in a comparable and objective manner, 500 random micrographs used for the final model determination were selected from each dataset. Particles were picked using cryoSPARC v4.6.0 blob picker job on denoised micrographs setting the minimum particle diameter to 50 Å and the maximum at 130 Å. The particles with a normalized cross-correlation (NCC) value higher than 0.8 and a power threshold value between 12 and 36 were extracted using 256 pixels box size. The particles were 2D classified and the classes where resolution was better than 10 Å were selected. The particles were 3D classified by heterogeneous refinement applying an initial low-pass resolution of 20 Å to the final WT closed conformation volume, the WT open conformation volume and four other models generated with junk particles from an independent dataset. The particles confidently assigned to the closed and the open conformation (3D class probability filter threshold 0.999) were independently 3D classified again using the same reference volumes. The final number of closed and open conformation particles were determined by the particles that were unambiguously assigned to the same conformation after the second 3D classification. The ratio was presented as the percentage of closed conformation particles with respect to the final number of closed and open conformation particles, in front of the percentage of open conformation particles.

## Model building

An initial model of MUC5AC-D3 Arg996Gln dimer was built in AlphaFold2 (Jumper et al, 2021). It was fitted into the density map with Molrep (Vagin and Teplyakov, 1997) and manually built using Coot 0.9.8.1. The model was refined along different iterations using the real-space refinement tool in Phenix 1.20.1-4487 (Liebschner et al, 2019) and manual refinement in Coot. The final structure refinement validation was performed by Molprobity (Williams et al,

2018). The other models were built following the same protocol but using the final MUC5AC-D3 Arg996Gln dimer model (WT and Arg996Gln Arg1201Trp dimers, and Arg996Gln tetramer) or the WT dimer model (Arg1201Trp) as initial models. PyMol (Schrodinger, LLC (2015) The PyMOL Molecular Graphics System, version 2.5) and UCSF ChimeraX (Meng et al, 2023) were used for structure analysis and figures generation.

The MUC5AC-D3 open conformation model was generated manually fitting the MUC5AC-D3 WT closed conformation C8-3 and TIL3 dimer and two independent VWD3 domains into the low-resolution density assisted by the fit in map tool ChimeraX. The model building was finished using ISOLDE (Croll, 2018).

### Human samples

Stomach biopsies were acquired during routine endoscopies at the Sahlgrenska University Hospital (ethical permission 085-06). All subjects gave written informed consent. Samples were fixed directly in Carnoy´s solution (composition 60% methanol, 30% chloroform, and 10% glacial acetic acid). Researchers had access to medical records.

### Immunofluorescent staining of histological sections

Sections were baked on the slides at 60 °C for 2 h, dewaxed using xylene and hydrated. Antigen heat-induced epitope retrieval was performed with 10 mM citrate buffer pH 6.0 at 100 °C for 20 min and then room temperature for another 20 min. Sections were washed in PBS, a barrier was drawn with an ImmEdge® Hydrophobic Barrier PAP Pen (H-4000, Vector laboratories, Newark, CA). Unspecific epitopes were blocked with 3% donkey serum in Tris-buffered saline and sections were permeabilized with 0.1% Triton X-100. Stainings were performed with sequential incubation with custom-made polyclonal rabbit anti-human MUC5B antibodies (1:200) in block solution (Fakih et al, 2020) overnight at 4 °C and monoclonal mouse anti-human MUC5AC (Lidell et al, 2008) (1:200) in block solution overnight at 4 °C (Cat# ab3649, Abcam, Cambridge, UK, RRID:AB_2146844). Secondary antibody was polyclonal donkey anti-rabbit Alexa Fluor 488 (Cat# A-21206, Thermo Fisher Scientific, Waltham, MA, RRID:AB_2535792) and polyclonal donkey anti-mouse Alexa Fluor 647 (Cat# A-31571, Thermo Fisher Scientific, RRID:AB_162542). Nuclei were stained with Hoechst 34580 (Thermo Fisher Scientific).

### Imaging

Images were acquired using ZEISS ZEN Microscopy Software (Carl Zeiss, Oberkochen, Germany, RRID:SCR_013672) on the ZEISS LSM900 with Airyscan 2 (Carl Zeiss, Oberkochen, Germany, RRID:SCR_022263) and processed with Imaris version 9.5 software (Oxford Instruments, Abingdon, UK, RRID:SCR_007370).

### Piglet airway tissue

Ethical permission for experiments involving newborn piglets (Sus scrofa domesticus) was obtained from Regierungen von Ober-bayern, Munich, Germany (AZ55.2-1-54-2531-78-07) and Jord-bruksverket, Jönköping, Sweden (Dnr 6.7.18-12708/2019).

Tracheas were acquired from wild-type piglets (Sus scrofa domesticus). To induce birth, intramuscular administration of 0.175 mg Cloprostenol (Estrumate®, Intervet GmbH, Unterschleis-sheim, Germany), on gestation day 112–114. Within 24 h of birth, piglets were anesthetized by Ketamine (Ursotamin®, Serumwerk Bernburg, Germany) and Azaperone (Stresnil®, Elanco Animal Health, Bad Homburg, Germany) and killed by intracardial injection of Tanax® T61 euthanasia solution (Intervet GmbH, Unterschleissheim, Germany). Tracheas from the larynx and lungs were excised and the lung parenchyma removed under Perfadex® solution, pH 7.2 (XVIVO Perfusion, Gothenburg, Sweden) before the prepared airways including larynx, trachea and bronchi were transferred to a 50 ml tube with Perfadex® solution, pH 7.2 and shipped at 4 °C overnight to Gothenburg.

### Electron microscopy

Distal tracheal tissue (two to three cartilage rings in length) from newborn piglets was fixed in modified Karnovsky's fixative (2% paraformaldehyde, 2.5% glutaraldehyde in 0.05 M sodium cacodylate buffer, pH 7.2) for 24 h at 4 °C. Postfixation was performed in 1% $OsO_4$ at 4 °C three times with intervening 1% thiocarbohydrazide steps. The samples were dehydrated with increasing concentrations of ethanol followed by hexamethyldisilazane that was allowed to evaporate. Samples were mounted on aluminum specimen pin stubs (Cat# AGG301, Agar Scientific, Stansted, Essex, UK) with carbon tabs (Cat# AGG3347N, Agar Scientific, Stansted, Essex, UK) and conductive silver paint (Cat# 16040-30, Ted Pella, Redding, CA). To decrease charging, samples were sputter-coated with palladium before imaging at 3 kV in a field emission scanning electron microscope (Zeiss DSM 982 Gemini, Carl Zeiss, Oberkochen, Germany).

## Data availability

The structural data from cryoEM is deposited at PDB and EMDB with accession www.rcsb.org/structure/8QTV and www.ebi.ac.uk/emdb/18654 (D'-D3-CysD1 WT), www.rcsb.org/structure/8QTB and www.ebi.ac.uk/emdb/18648 (D'-D3-CysD1 Arg996Gln), www.rcsb.org/structure/8R1U and www.ebi.ac.uk/emdb/18828 (D3 Arg1201Trp), www.rcsb.org/structure/8R1Z and www.ebi.ac.uk/emdb/18829 (D3 Arg996Gln Arg1201Trp) and www.rcsb.org/structure/8QSP and www.ebi.ac.uk/emdb/18638 (6xHis-D3 Arg996Gln).

The source data of this paper are collected in the following database record: biostudies:S-SCDT-10_1038-S44319-025-00395-8.

## Peer review information

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

## Acknowledgements

This work was supported by the The Knut and Alice Wallenberg Foundation (2017.0028), European Research Council ERC (101100663, 694181), Swedish Research Council (2017-00958, 2022-00646), IngaBritt and Arne Lundberg Foundation (2018-0117), Sahlgren's University Hospital (ALFGBG-440741), Bill and Melinda Gates Foundation (OPP1202459), The Swedish Heart-Lung Foundation (20220404 (AE); 20230413 (GCH)), and Wilhelm and Martina Lundgren's Foundation. We are indebted to the Mammalian Protein Expression Core facility at University of Gothenburg for help with protein expression. The data was collected at the cryoEM Swedish National Facility funded by the Knut and Alice Wallenberg, Family Erling Persson and Kempe Foundations, SciLifeLab, Stockholm University and Umeå University.

## Author contributions

**Sergio Trillo-Muyo**: Conceptualization; Data curation; Formal analysis; Validation; Investigation; Visualization; Methodology; Writing—original draft; Writing—review and editing. **Anna Ermund**: Conceptualization; Data curation; Investigation; Writing—review and editing. **Gunnar C Hansson**: Conceptualization; Resources; Data curation; Supervision; Funding acquisition; Validation; Visualization; Writing—original draft; Project administration; Writing—review and editing.

Source data underlying figure panels in this paper may have individual authorship assigned. Where available, figure panel/source data authorship is listed in the following database record: biostudies:S-SCDT-10_1038-S44319-025-00395-8.

## Funding

## Disclosure and competing interests statement

The authors declare no competing interests.

