## [Peer Review File · EMBO Reports]

Structure of the MUC5AC VWD3 assembly responsible for the formation of net-like mucin polymers

Sergio Trillo-Muyo, Anna Ermund, and Gunnar Hansson

Corresponding author(s): Gunnar Hansson (gunnar.hansson@medkem.gu.se), Sergio Trillo-Muyo (sergio.muyo@medkem.gu.se)

Review Timeline:

Transfer Date:	11th Oct 24
Editorial Decision:	16th Oct 24
Revision Received:	12th Nov 24
Editorial Decision:	12th Dec 24
Revision Received:	3rd Jan 25
Accepted:	21st Jan 25

Transaction Report: This manuscript was transferred to EMBO reports following peer review at The EMBO Journal.

Referee #1:

In this study Trillo-Muyo et al. use single particle cryoEM and molecular modeling to generate a structural model of the D3 domain of human mucin MUC5AC. In building their model, they took advantage of similar domains found in other proteins, the VonWillebrand Factor and mucin MUC2, and were able to compare and contrast the D3 domain across the three proteins. Through their model, they identified cysteines involved in MUC5AC's covalent N-terminal dimerization, and proposed that the unique elements of MUC5AC's TIL3 domain permit MUC5AC's N-terminal dimer to non-covalently organize into tetramers. Their cryoEM classification revealed a set of particles that were larger than the others and they attribute this to an "open conformation" of MUC5AC that may have a role in biophysical properties and interactions of the molecule. Finally, they also made structural models of two SNPs (Arg996Gln and Arg1201Trp) and compared those models to the wild type model. They present GWAS data suggesting that Arg1201Trp is more prevalent in individuals with chronic obstructive pulmonary disease (COPD) and idiopathic pulmonary fibrosis (IPF).

Comments

Overall the first part of the work presented here: the structural model of MUC5AC's D3 domain, descriptions for MUC5AC's N-terminal multimerization and the potential for an "open conformation" are of great value to the community and the methodology used to produce the model seemed fundamentally sound. I would caution that the model presented may or may not fully capture the entire N-terminal region's structural organization or functional properties, and this point should be made clearer in the manuscript.

The connection between MUC5AC SNPs and disease, however, is weakly supported. The p-value cutoffs used to determine significance of the differences in SNP frequency in control and diseased population are high and this will likely produce false positives. No data is presented showing significant structural differences in the SNP variants, and the attempt to causally connect the Arg1201Trp SNP to disease is highly speculative.

Major Points

1) The positive disease correlation between the SNP rs878913005 and COPD and IPF is weakly supported. In GWAS using a p-value significance threshold of $< .05$ will yield a huge number of false positives due to the size of the data set. Furthermore, it is important to also use mixed variable in multiple models to validate their findings and consider co-variables, such as the study population, age, race, gender, smoking status, bmi sites, familial relationships, etc..

2- In the discussion the authors speculate on a causal role between the SNP rs878913005 and disease. Attributing a causal role to a single SNP is extremely difficult due to clustering. The authors didn't observe significant conformational differences between the rs878913005 variant and the wild type and suggest that instead the variant affects molecular dynamics. Extrapolating from this point to the behavior of the full length molecule and its functional role in disease is too far of a reach.

Given these major issues with this part, this reviewer would strongly recommend that the authors consider removing the SNP analysis from the manuscript. Without more rigorous statistical validation and supporting experimental data, this part of the study undermines the overall strength of the paper.

Minor:

1. Typo in the second paragraph of the introduction, "granulke"
2. In the existing MUC5AC literature, TIL3 is a domain before the D3 domain. I understand from the intro and VWF literature that each D domain is composed of a D,C-8,TIL, and E domain. Is there an easy way to explain this to readers?
3. Typos in the third paragraph "contains" and "interacting"
4. Typo last paragraph of the intro. "way they differ as" -> "way they differ has"
5. D1 and D2 are not cleaved in MUC5B and MUC5AC, and MUC5AC has additional differences in the D1 domain. In the MUC5AC-N covalent dimer section of the results, the sentence suggesting that D1 and D2 are of less importance should be removed.
6. The absence of D1 and D2 in this construct should be acknowledged as a weakness of the model.
7. Are all dimers either "closed" or "open"? Can a mixed state exist with one subunit open and another closed? If not observed could that be a consequence of constraints from the symmetry imposed during reconstruction?
8. To support the observation that the ratio of open vs closed states changes with the variants, it would be useful to present the ratio of particles in the open and closed configurations for the WT, as well as the variants.
9. Figures 5A and 5B are unsupported and probably should be removed, there is no supporting evidence for the specific angles of 40 degrees and 180 degrees. The tetrameric interaction was said to be flexible and the tetramer structure was solved using a structure that only appeared in the variant + his-tag. This shouldn't be the basis for a generalization regarding full length MUC5AC organization.
10. In the linkage disequilibrium results, what do the values represent? R2 or D'?

Referee #2:

General impressions:

The paper employs current methods of Cryo EM to elucidate the detailed structure of expressed, important N-terminal VWD domains of the secreted respiratory MUC5AC. The authors speculate about the roles of these structures in mucin packaging, polymerization and assembly. They also examine the effects of various point mutations on the structure of mucin and investigate the presence of these mutations in selected populations and their associations with certain diseases.

There remain some outstanding questions or items that would benefit from further clarification.

Figure 1 is very clear and informative and clearly explained

The understanding of the descriptions on figure 2C would be enhanced by the inclusion of 'black arrows' as in figure 2B, to indicate the differences in the structures of MUC5AC loops as compared to MUC2 and VFW loops. The discussion of the structural differences in the MUC5AC-D3 TIL3 structures should include the significance of these differences.

Positing the existence and structure of the 'open form' based on particles of 'low resolution' with severe preferred orientation problem' is somewhat problematic and speculative. Can this assignment be justified quantitatively?

The authors state that the N-terminal of VWD and C-terminal of C8 point in the same direction and are only 12A apart. Would this also be the case in the total intact MUC5AC molecule where other domains would be attached to these N and C termini?

"In the C8-3 domain interaction region MUC5AC-D3 presents a phenylalanine (Phe1086, Figure 1E) instead of a histidine (MUC2-D3 His1042), potentially affecting the effect of pH in the intracellular packing." (Page 4). The differences between MUC2 and MUC5AC in terms of the pH-responsiveness of their intracellular packing could be elaborated on.

The authors state that the structures of the MUC5AC-D3 R996Q and R1201W variants exhibit little difference from the WT structure. More explanation needs to be given as to how these insignificant structural differences relate to or account for the pathologies observed in the discussed COPD and IFP disorders. Moreover, the authors' argument that these mutations affect the equilibrium between the open and closed forms is poorly supported given the sparse characterization of the open form and that the claim is based solely on a perceived difference in frequencies of the two forms during CryoEM instead of biochemical characterization.

The significance of the contribution of the HisTag HisTag-induced associations and ring structures seem unrelated to the basic structure and function of the VWD3 domains. It would be helpful to understand how the observed HisTag interactions affect the interpretation of their results.

The speculation for the existence of 'net-like' MUC5AC structures need experimental verification. One might expect that such extended, highly structured, regular lattice forms should be detectable or observed by other structural techniques such as AFM, EM, CryoEM, neutron or x-ray scattering.

Minor typographical issues:

Page 3, line 4 MUCB should be MUC5B

Page 11, line 6 formes should be forms

Page25, line 17 starts should be stars.

Referee #3:

This is an intriguing manuscript that focusses on structural analysis that pertains to how the respiratory mucin MUC5AC forms higher-order structures (covalent dimers and non-covalent tetramers and octamers), how two SNPs influence the structure, the associations of the SNPs with COPD and IPF and how these putative changes in MUC5AC might impact on mucus in these conditions. The work is original, and findings are interesting and provide insight into how the crosslinked networks of MUC5AC reported in other studies (e.g. Carpenter et al., 2021, PNAS). While the structural analyses presented are novel and appear generally sound, the association of the SNPs to COPD and IPF is based on rather weak genetic analysis and consequently the discussion of their effects on the two respiratory conditions is very speculative. This reviewer has several concerns relating to the manuscript in its current form.

Structural analyses

The study was performed on only a portion of the region of the mucin involved in formation of the covalent dimers - the MUC5AC-D3. While the authors clearly show biochemically that this region of the mucin forms disulfide-stabilized dimers, can they be sure that the other regions of the MUC5AC N-terminus play no role in how the full-length mucin forms covalent dimers? The MUC5AC-N covalent dimer structure is shown in the reduced form (which is claimed to be a partial variant of MUC5AC-N), why not show the oxidised form (or both)? The cryoEM suggests that the MUC5AC-N covalent dimer can form tetramers (in different conformations) and the R966Q octamers. The proportions of these different forms are hard to gauge from the results and is the strength of the interactions. It would be important to show that these different oligomers can form by another technique, e.g., in solution, and this could be achieved by using the gel filtration approach used in the purification of the different MUC5AC-N variants.

In figure 5, a schematic representation of the ideal structure of the MUC5AC non-covalently cross-linked network is shown and this is claimed to resemble the structure of pig lung MUC5AC network. The resolution of the images for the fixed stomach tissue and the scanning EM is too low to draw any conclusions on the structure of the MUC5AC network and so adds little value to the results. Moreover, as they state in the text these networks might be generated by covalent crosslinks between the CysD-domains of MUC5AC as this group has previously shown for MUC2 in intestinal mucus. I wonder if they have considered isolating MUC5AC directly from mucin granules to show the network structure of MUC5AC and if it can be taken apart with high-salt and/or detergents? This would greatly improve this aspect of the manuscript.

Genetic analysis

The manuscript reports the potential impact of two MUC5AC SNPs on MUC5AC structure and how this may have relevance for COPD and IPF. However, the analyses performed to show this are quite limited compared with recent publications assessing associations of SNPs in mucin genes with respiratory conditions (e.g., Altman et al., 2021, *J Allergy Clin Immunol*. Shrine et al., 2019, *Lancet Respir Med*). I am surprised that there is no mention of any co-variants (such as sex, age, FEV) used in the analysis. Also, the novel findings reported here need to be replicated in other cohorts. While the findings are potentially very important, my work needs to be done here.

Referee #4:

Sergio T. et al., report a series reconstruction of gel-forming mucins MUC5AC, and based on structural analysis combined with other approaches such as SEM and immunofluorescent staining to demonstrate the formation of net-like architecture. The authors first examined multiple structures from different constructs over MUC5AC protein, compared their structures and demonstrated how different mutations affect their structures. Then by analyzing data from patients, authors proposed a possible role for MUC5AC in lung disease, such as IPF.

Major comments

1. In title and abstract, SNP were mentioned several times, but what's this abbreviation for? The authors should provide full name, since it's important to understand the story.
2. In Fig 1D, authors described the existence of calcium ions in their model, is it sufficient to see the existence of calcium ions at a resolution around 3 angstroms? The approaches of how calcium ions were identified should be described, otherwise this figure legends as well as the figure should be changed.
3. The information over closed-open conformation equilibrium of MUC5AC is vital to this study, yet how the 3D map on open conformation showed in Fig 3 were poorly described in Method section, and there is no information regarding how these maps were generated in Supplementary figures. Furthermore, information regarding resolution, modeling quality etc. of the open conformation maps was also not provided throughout the description in Method, neither in Supplementary figures.
4. Related to previous comments, in Fig 3D, authors described a putative salt bridge and a hydrogen bond, firstly, these statements should be valid only after modeling quality has been shown, secondly, distance measurement over these bonding residues should be provided on the figures, to make a better understanding over these interaction sites.
5. In Fig. 5A authors described the tetrameric assembly of MUC5AC-D3 Arg996Gln maps, with protruding PTS domains, it is confusing whether the densities of PTS domains were actually solved in the cryo-EM map or not? If that is actual density, in Supplementary Fig 2 authors should show such density, if that is a proposed formation, authors should make a difference in showing actual density maps than PTS domain in both Fig. 5A and 5B, for better clarification.
6. Related to previous comment, if the formation of PTS domain was just a proposed conformation, additional information over how 40-degree angle is defined should be stated in the literature.
7. Clearly the statement of 'Fig S1 to S5' in Page 14, line 22 is wrong, since Fig S2 contains only local resolution estimation result, and Fig S6 also contains data processing steps. Authors should carefully check their statement about supplementary figures throughout this manuscript.
8. In Supplementary Figures 1, 3, 4, 5 and 6, the particle orientation distribution exhibit preferred orientation to certain extent, 3D FSC estimation should also be supplied for these 5 maps, to indicate whether preferred orientation distribution affected map density or not.
9. In Supplementary Figures, detailed map-model fitting quality should be indicated, at least use some representative regions such as dimerization interface to show the modeling quality over these

cryo-EM maps. Also, in Supplementary Table S1, the resolution value of map-model FSC at 0.5 were provided, these FSC curves should also be supplied along with the demonstrations over map-model fitting quality.

Minor comments

1. Page 8, line 16, '...chromosome 11 with 39,853 nucleotides in between (Figure 5F)', lacking reference information.
2. Page 14, line 4, authors should indicate which equipment they used to glow discharge their grids.
3. Page 14, line 8, it is unclear what the authors meant by 'dose per image'. Is it per frame or per movie?
4. Page 14, line 9, 'in 0.3 steps' lacking the unit of length.
5. Page 14, line 19, 'CTF correction' should be 'CTF parameters estimation' or 'defocus estimation' since correction of CTF effect doesn't happen at this step.
6. Page 14, 15 and 16 lacking some essential information when describe the detailed data processing regarding each construct, such as which software was used for manual particle picking, for blob-based particle picking, for 2D classification etc. these information should be stated, and also the references should be provided.
7. Naming convention should be unified throughout literature. For instance, the names on each map described in Supplementary Table are different from the ones in Supplementary Figures.

Recommendation

The authors provided new insight regarding how MUC5AC form the mucus. A combination of biochemistry, structural and other methods provide evidences supporting their claims, still, some of the maps and models presented in the manuscript lacking essential information, such as resolution estimation for open conformation, 3D FSC for all the reconstructed maps, modeling quality assessment etc. without these information, some of the statements made in the manuscript seem to be over claimed. Hence, I would suggest to consider this manuscript to be published after major revisions, also refinement on literatures have been done.

Dear Gunnar,

Thank you for the transfer of your research manuscript to our journal. As my colleague Ieva Gailite from The EMBO Journal told you, we are interested in considering your manuscript for publication after a revision. Please focus on the structural analysis of WT MUC5AC and address all referee concerns related to these data. The data on the SNP variants should be removed, as suggested by three of the referees. You already indicated that you agree with this proposal and that you are currently revising your manuscript along these lines. Depending on the nature of the revisions and changes, your manuscript will be seen again by the referees, in particular referee #4.

You indicated that the revision and removal of the SNP part will result in authorship changes. Upon resubmission, please provide written consent by all authors regarding the change in authorship.

Please address all referee concerns in a complete point-by-point response. Acceptance of the manuscript will depend on a positive outcome of a second round of review. It is EMBO Reports policy to allow a single round of revision only and acceptance or rejection of the manuscript will therefore depend on the completeness of your responses included in the next, final version of the manuscript.

I am also happy to discuss the revision further via e-mail or a video call, if you wish.

Your manuscript will be published as short reports. For short reports, the revised manuscript should not exceed 27,000 characters (including spaces but excluding materials & methods and references) and 5 main plus 5 expanded view figures. The results and discussion sections must further be combined, which will help to shorten the manuscript text by eliminating some redundancy that is inevitable when discussing the same experiments twice.

I list the general formatting guidelines below but would like to point out a few required changes:

- The Data availability section needs links that resolve directly to the datasets.
- The Author contributions must be removed from the manuscript. The information entered in the online manuscript tracking system will be retrieved and typeset into the article. Please make sure that all information is up-to-date.
- Declaration of Interest should be called "Disclosure and competing interests statement"
- Figure legends should state the exact p-values.

2) individual production quality figure files as .eps, .tif, .jpg (one file per figure). Please download our Figure Preparation Guidelines (figure preparation pdf) from our Author Guidelines pages <https://www.embopress.org/page/journal/14693178/authorguide> for more info on how to prepare your figures.

4) a complete author checklist, which you can download from our author guidelines (<https://www.embopress.org/page/journal/14693178/authorguide>). Please insert information in the checklist that is also reflected in the manuscript. The completed author checklist will also be part of the RPF.

5) Please note that all corresponding authors are required to supply an ORCID ID for their name upon submission of a revised manuscript (<https://orcid.org/>). Please find instructions on how to link your ORCID ID to your account in our manuscript tracking system in our Author guidelines (<https://www.embopress.org/page/journal/14693178/authorguide#authorshipguidelines>)

6) We replaced Supplementary Information with Expanded View (EV) Figures and Tables that are collapsible/expandable online. A maximum of 5 EV Figures can be typeset. EV Figures should be cited as 'Figure EV1, Figure EV2' etc... in the text and their respective legends should be included in the main text after the legends of regular figures.

- For the figures that you do NOT wish to display as Expanded View figures, they should be bundled together with their legends in a single PDF file called *Appendix*, which should start with a short Table of Content. Appendix figures should be referred to in

the main text as: "Appendix Figure S1, Appendix Figure S2" etc. See detailed instructions regarding expanded view here: <<https://www.embopress.org/page/journal/14693178/authorguide#expandedview>>

7) The Data availability section needs links that resolve directly to the deposited datasets. Suggested wording: "The [structural coordinates | microarray | mass spectrometry] data from this publication have been deposited to the [name of the database] database [URL] and assigned the identifier [accession | permalink | hashtag].". Should this not apply, this should still be stated as "This study includes no data deposited in external repositories."

Additional information on source data and instruction on how to label the files are available <<https://www.embopress.org/page/journal/14693178/authorguide#sourcedata>>.

10) Figure legends and data quantification:
The following points must be specified in each figure legend:

- the name of the statistical test used to generate error bars and P values,
 - the number (n) of independent experiments (please specify technical or biological replicates) underlying each data point,
 - the nature of the bars and error bars (s.d., s.e.m.)
- If the data are obtained from n {less than or equal to} 5, show the individual data points in addition to the SD or SEM.
- If the data are obtained from n {less than or equal to} 2, use scatter blots showing the individual data points.

See also the guidelines for figure legend preparation:
<https://www.embopress.org/page/journal/14693178/authorguide#figureformat>

11) Our journal encourages inclusion of *data citations in the reference list* to directly cite datasets that were re-used and obtained from public databases. Data citations in the article text are distinct from normal bibliographical citations and should directly link to the database records from which the data can be accessed. In the main text, data citations are formatted as follows: "Data ref: Smith et al, 2001" or "Data ref: NCBI Sequence Read Archive PRJNA342805, 2017". In the Reference list, data citations must be labeled with "[DATASET]". A data reference must provide the database name, accession number/identifiers and a resolvable link to the landing page from which the data can be accessed at the end of the reference. Further instructions are available at <<https://www.embopress.org/page/journal/14693178/authorguide#referencesformat>>.

12) All Materials and Methods need to be described in the main text using our 'Structured Methods' format. According to this format, the Methods section includes a Reagents and Tools Table (listing key reagents, experimental models, software and relevant equipment and including their sources and relevant identifiers) followed by a Methods and Protocols section describing the methods, ideally using a step-by-step protocol format. The aim is to facilitate adoption of the methodologies across labs. Please download and fill our Reagents and Tools Table template (.docx), which you can find in our author guidelines: <https://www.embopress.org/page/journal/14693178/authorguide#structuredmethods>. When submitting your revised manuscript, please do not include the Reagents and Tools Table in the Methods section of the manuscript but upload it as a separate file choosing the file type "Reagent Table". An example of a Method paper with Structured Methods can be found here: <https://www.embopress.org/doi/10.15252/msb.20178071>.

13) As part of the EMBO publication's Transparent Editorial Process, EMBO Reports publishes online a Review Process File to accompany accepted manuscripts. This File will be published in conjunction with your paper and will include the referee reports, your point-by-point response and all pertinent correspondence relating to the manuscript.

Kind regards,

Martina

COMMENTS TO REFEREES

Referee #1:

In this study Trillo-Muyo et al. use single particle cryoEM and molecular modeling to generate a structural model of the D3 domain of human mucin MUC5AC. In building their model, they took advantage of similar domains found in other proteins, the VonWillebrand Factor and mucin MUC2, and were able to compare and contrast the D3 domain across the three proteins. Through their model, they identified cysteines involved in MUC5AC's covalent N-terminal dimerization, and proposed that the unique elements of MUC5AC's TIL3 domain permit MUC5AC's N-terminal dimer to non-covalently organize into tetramers. Their cryoEM classification revealed a set of particles that were larger than the others and they attribute this to an "open conformation" of MUC5AC that may have a role in biophysical properties and interactions of the molecule. Finally, they also made structural models of two SNPs (Arg996Gln and Arg1201Trp) and compared those models to the wild type model. They present GWAS data suggesting that Arg1201Trp is more prevalent in individuals with chronic obstructive pulmonary disease (COPD) and idiopathic pulmonary fibrosis (IPF).

Comments

Overall the first part of the work presented here: the structural model of MUC5AC's D3 domain, descriptions for MUC5AC's N-terminal multimerization and the potential for an "open conformation" are of great value to the community and the methodology used to produce the model seemed fundamentally sound. I would caution that the model presented may or may not fully capture the entire N-terminal region's structural organization or functional properties, and this point should be made clearer in the manuscript.

The connection between MUC5AC SNPs and disease, however, is weakly supported. The p-value cutoffs used to determine significance of the differences in SNP frequency in control and diseased population are high and this will likely produce false positives. No data is presented showing significant structural differences in the SNP variants, and the attempt to causally connect the Arg1201Trp SNP to disease is highly speculative.

ANSWER: Thank you for the positive comments and as suggested, we have now removed the SNP analyses in relation to disease.

Major Points

1) The positive disease correlation between the SNP rs878913005 and COPD and IPF is weakly supported. In GWAS using a p-value significance threshold of $< .05$ will yield a huge number of false positives due to the size of the data set. Furthermore, it is important to also use mixed variable in multiple models to validate their findings and consider co-variables, such as the study population, age, race, gender, smoking status, bmi sites, familial relationships, etc..

ANSWER: We have now removed the SNP analyses in relation to disease.

2- In the discussion the authors speculate on a causal role between the SNP rs878913005 and disease. Attributing a causal role to a single SNP is extremely difficult due to clustering. The authors didn't observe significant conformational differences between the rs878913005

variant and the wild type and suggest that instead the variant affects molecular dynamics. Extrapolating from this point to the behavior of the full length molecule and its functional role in disease is too far of a reach.

Given these major issues with this part, this reviewer would strongly recommend that the authors consider removing the SNP analysis from the manuscript. Without more rigorous statistical validation and supporting experimental data, this part of the study undermines the overall strength of the paper.

ANSWER: We have followed the recommendation and removed the SNP analyses in relation to disease.

Minor:

1. Typo in the second paragraph of the introduction, "granulke"

ANSWER: Corrected

2. In the existing MUC5AC literature, TIL3 is a domain before the D3 domain. I understand from the intro and VWF literature that each D domain is composed of a D,C-8,TIL, and E domain. Is there an easy way to explain this to readers?

ANSWER: We think the literature is wrong or the reviewer has misunderstood the order of the domains and that the VWD domain is a part of a VWD assembly. The 3.5 VWD assemblies of the mucins can of course be numbered indifferent ways, but as the first assembly starts with a VWD domain and the third ends with a TIL and E, it is most logic to number them according to this.

3. Typos in the third paragraph "contains" and "interacting"

ANSWER: Corrected.

4. Typo last paragraph of the intro. "way they differ as" -> "way they differ has"

ANSWER: Corrected

5. D1 and D2 are not cleaved in MUC5B and MUC5AC, and MUC5AC has additional differences in the D1 domain. In the MUC5AC-N covalent dimer section of the results, the sentence suggesting that D1 and D2 are of less importance should be removed.

ANSWER: We have removed this comment, the sentence now reads: 'The VWD1-VWD2 (D1-D2) assemblies of VWF and MUC2 are cleaved off after polymerization suggesting that these domains are primary required for the granule packing.'

6. The absence of D1 and D2 in this construct should be acknowledged as a weakness of the model.

ANSWER: We and the group in Dublin (Ryan et al) has spent years on trying to obtain the MUC5AC N-terminus. The protein is produced in CHO and HEK cells, but accumulates in the endoplasmic reticulum and does not pass into Golgi. This accumulation sometimes causes the cells to rupture and immature MUC5AC-N can be found in the culture medium as for example in the Carpenter et al. That this is the case was observed by all N-glycans unprocessed as high-mannose N-glycans. That the MUC5AC mucin is different from the other mucins suggests an additional level of regulation of expression/secretion that has not been explored.

We fully agree that this is a weakness and we altered the last sentence of the first paragraph to: 'Unable to produce the mature, secreted MUC2-N, we had to focus on the MUC5AC-D3 (Figure 1A and B) assembly and its interactions.'

7. Are all dimers either "closed" or "open"? Can a mixed state exist with one subunit open and another closed? If not observed could that be a consequence of constraints from the symmetry imposed during reconstruction?

ANSWER: No, we could not observe a mixed state. There are 2D classes in the open and in the closed conformation where only one monomer is well defined and the other is blurry or faint suggesting flexibility and heterogeneity, but we could not observe a clear mixed state. The two states were initially identified analyzing 2D classifications where no symmetry is imposed and we could not observe a mixed state. The symmetry constrains were only applied in the last refinement steps as they improved the model quality, but they did not affect the 2D and 3D particle classification or the deep learning particle picking training. If the mixed state exist it is less abundant than the closed and the open states, suggesting that the state transition is allosteric.

8. To support the observation that the ratio of open vs closed states changes with the variants, it would be useful to present the ratio of particles in the open and closed configurations for the WT, as well as the variants.

ANSWER: We completely agree with this comment. We applied and standardized blob picking method to all the datasets using the same parameters in order to be able to compare the results. We have added the calculated ratios and updated the materials and methods.

9. Figures 5A and 5B are unsupported and probably should be removed, there is no supporting evidence for the specific angles of 40 degrees and 180 degrees. The tetrameric interaction was said to be flexible and the tetramer structure was solved using a structure that only appeared in the variant + his-tag. This shouldn't be the basis for a generalization regarding full length MUC5AC organization.

ANSWER: We agree that this is speculative, but we would like to keep these as it helps the reader to understand the importance of the interactions within the D3 assembly and why a laminated structure that is different from the linear MUC5B mucin (Fig. 5CD) is obtained.

10. In the linkage disequilibrium results, what do the values represent? R2 or D'?

ANSWER: Sorry for the mistake, D', was removed.

Referee #2:

General impressions:

The paper employs current methods of Cryo EM to elucidate the detailed structure of expressed, important N-terminal VWD domains of the secreted respiratory MUC5AC. The authors speculate about the roles of these structures in mucin packaging, polymerization and assembly. They also examine the effects of various point mutations on the structure of mucin and investigate the presence of these mutations in selected populations and their associations with certain diseases.

There remain some outstanding questions or items that would benefit from further clarification.

Figure 1 is very clear and informative and clearly explained

ANSWER: Thank you.

The understanding of the descriptions on figure 2C would be enhanced by the inclusion of 'black arrows' as in figure 2B, to indicate the differences in the structures of MUC5AC loops as compared to MUC2 and VFW loops. The discussion of the structural differences in the MUC5AC-D3 TIL3 structures should include the significance of these differences.

ANSWER: An arrow has been added to Figure 2C and the figure text amended.

Positing the existence and structure of the 'open form' based on particles of 'low resolution' with severe preferred orientation problem' is somewhat problematic and speculative. Can this assignment be justified quantitatively?

ANSWER: We based the existence of an open form on the high quality 2D classes where we can distinguish the VWD and the C8 domains and their relative orientation, but we agree that a quantitatively approach would be beneficial. We have made the calculations as discussed for review 1 point 8.

The authors state that the N-terminal of VWD and C-terminal of C8 point in the same direction and are only 12Å apart. Would this also be the case in the total intact MUC5AC molecule where other domains would be attached to these N and C termini?

ANSWER: We used both the D'-D3-CysD1 domains and the isolated D3 assembly for cryoEM structure determination and we observed that the presence of the contiguous domains does not affect the relative position of the N-terminal of VWD and C-terminal of C8. We could observe how their relative position changes between the open and closed conformation. The presence of the D1 and D2 domain in the N-terminal part is not expected to have any impact as these are flexible and do not interact with D3 at pH 7.4, but the huge mucin domain and the rest of the domains in the C-terminal part could have an effect in the open and closed conformation equilibrium. The polymer formation in the N-terminal part of the mucin is driven by the disulfide bonds in the C8-3 and TIL3 domain, so the shearing forces and tension should not be transmitted to the interface between C8-3 and VWD3 but could affect the interaction between VWD3 and TIL3. Unfortunately, it is not possible to structurally study full MUC5AC molecules to the detail required to determine this point.

"In the C8-3 domain interaction region MUC5AC-D3 presents a phenylalanine (Phe1086, Figure 1E) instead of a histidine (MUC2-D3 His1042), potentially affecting the effect of pH in the intracellular packing." (Page 4). The differences between MUC2 and MUC5AC in terms of the pH-responsiveness of their intracellular packing could be elaborated on.

ANSWER: To our knowledge, nothing is described about differences between MUC2 and MUC5AC in terms of pH-responsiveness. We just suggested that the difference could have a potential role. The intracellular packing at low pH of both molecules seems to be different, this could be part of the explanation even though it is likely more complex. We focused the present work on extracellular conditions, so we prefer not to elaborate more on this part.

The authors state that the structures of the MUC5AC-D3 R996Q and R1201W variants exhibit little difference from the WT structure. More explanation needs to be given as to how these insignificant structural differences relate to or account for the pathologies observed in the discussed COPD and IFP disorders. Moreover, the authors' argument that these mutations affect the equilibrium between the open and closed forms is poorly supported given the sparse characterization of the open form and that the claim is based solely on a perceived difference in frequencies of the two forms during CryoEM instead of biochemical

characterization.

ANSWER: We fully agree with this comment. We applied and standardized blob picking protocol to all the datasets using the same parameters in order to be able to determine the ratios between the closed and the open conformation and compare the results. The perceived differences are now expressed as objective ratios. We have removed the association between SNPs and diseases.

New text has been added: ‘The same standardized method used in the WT dataset was used to calculate the ratio between the closed and open conformation in all the variants. The ratio in Arg996Gln dataset, 98:2, illustrates the important reduction in open conformation particles from 17% in the WT to only 2%. The Arg1201Trp mutation implied a more discrete reduction in open conformation particles showing a ratio of 89:11 between the closed and open conformation. However, it represents a reduction of 35% respect the WT. The double mutant Arg996Gln Arg1201Trp showed a ratio of 96:4 confirming the effect of the mutations’

The significance of the contribution of the HisTag HisTag-induced associations and ring structures seem unrelated to the basic structure and function of the VWD3 domains. It would be helpful to understand how the observed HisTag interactions affect the interpretation of their results.

ANSWER: The results were validated using HisTag free D3 assemblies or the presence of the D’ domain disrupting the interactions. We agree that ring structures seem unrelated to the basic structure and function of the VWD3 domains, this is why we focused on the tetramers that could be observed in preparations without HisTag. Anyhow, upon a tetramer formation, there are no steric impediments for the chains not involved in the interaction to further interact with other D3 assemblies creating bigger oligomers. The HisTag-HisTag interactions clearly increase the oligomers stability. It could allow the observation of transient structures but could also introduce artifacts, this is why we chose not to overinterpret the data and just describe the structures that could be observed without HisTag using this data to obtain the high-resolution information.

The speculation for the existence of 'net-like' MUC5AC structures need experimental verification. One might expect that such extended, highly structured, regular lattice forms should be detectable or observed by other structural techniques such as AFM, EM, CryoEM, neutron or x-ray scattering.

ANSWER: The ‘net-like’ structures illustrated in Fig 5B is of course idealized and the net can never be expected to be ideal or fully extended. Illustrations supporting the ‘net-like’ MUC5AC is found in Fig. 5 CD. The way that a linear molecule can form laminated sheets as shown in the figure is by forming ‘flat’ structures (Fig. 5C). That the MUC5B form linear and the MUC5AC more complex ‘net-like’ structures are also illustrated in the EM Fig. 5D. The best EM of net-like structures is shown in the reference Carpenter et al. In this case, they removed either the MUC5B or MUC5AC genes from Calu3 cells. In the case of MUC5AC the figure show net-like structures.

Figure removed

(Fig 1B in Carpenter et al. (2021) PNAS 118, e2104490118)

Minor typographical issues:

Page 3, line 4 MUCB should be MUC5B

ANSWER: Corrected

Page 11, line 6 formes should be forms:

ANSWER: Corrected

Page 25, line 17 starts should be stars.

ANSWER: Corrected

Referee #3:

This is an intriguing manuscript that focusses on structural analysis that pertains to how the respiratory mucin MUC5AC forms higher-order structures (covalent dimers and non-covalent tetramers and octamers), how two SNPs influence the structure, the associations of the SNPs with COPD and IPF and how these putative changes in MUC5AC might impact on mucus in these conditions. The work is original, and findings are interesting and provide insight into how the crosslinked networks of MUC5AC reported in other studies (e.g. Carpenter et al., 2021, PNAS). While the structural analyses presented are novel and appear generally sound, the association of the SNPs to COPD and IPF is based on rather weak genetic analysis and consequently the discussion of their effects on the two respiratory conditions is very speculative. This reviewer has several concerns relating to the manuscript in its current form.

ANSWER: We have removed the SNP correlation with COPD and IPF from the current manuscript.

Structural analyses

The study was performed on only a portion of the region of the mucin involved in formation of the covalent dimers - the MUC5AC-D3. While the authors clearly show biochemically that this region of the mucin forms disulfide-stabilized dimers, can they be sure that the other regions of the MUC5AC N-terminus play no role in how the full-length mucin forms covalent dimers? The MUC5AC-N covalent dimer structure is shown in the reduced form (which is claimed to be a partial variant of MUC5AC-N), why not show the oxidised form (or both)?

ANSWER: As discussed in the manuscript and above to referee #1, we and others cannot produce the complete recombinant MUC5AC-N. The three MUC5AC-D3 recombinant proteins are shown both oxidized and reduced in Fig. 1C. All three migrate as apparent dimers.

The cryoEM suggests that the MUC5AC-N covalent dimer can form tetramers (in different conformations) and the R966Q octamers. The proportions of these different forms are hard to gauge from the results and is the strength of the interactions. It would be important to show that these different oligomers can form by another technique, e.g., in solution, and this could be achieved by using the gel filtration approach used in the purification of the different MUC5AC-N variants.

ANSWER: We observed the oligomers using other techniques but none of them reflected the mixture of states observed by cryoEM. The R966Q octamers represents around 80% of the particles observed in cryoEM, but in gel filtration they eluted in a single peak (see figure, Thyroglobulin: T, 669 kDa. Ferritin: F, 440 kDa. Aldolase: A, 158 kDa) between the expected elution volume for the octamer and the tetramer. All the other variants migrated as

single picks bigger than the dimer but smaller than the tetramer. We decided not include the data as it do not provide any insight on proportions or interaction strength.

Figure for referees not shown.

In figure 5, a schematic representation of the ideal structure of the MUC5AC non-covalently cross-linked network is shown and this is claimed to resemble the structure of pig lung MUC5AC network. The resolution of the images for the fixed stomach tissue and the scanning EM is too low to draw any conclusions on the structure of the MUC5AC network and so adds little value to the results. Moreover, as they state in the text these networks might be generated by covalent crosslinks between the CysD-domains of MUC5AC as this group has previously shown for MUC2 in intestinal mucus. I wonder if they have considered isolating MUC5AC directly from mucin granules to show the network structure of MUC5AC and if it can be taken apart with high-salt and/or detergents? This would greatly improve this aspect of the manuscript.

ANSWER: This question has been discussed above for referee #2. Especially Carpenter et al have as referred to addressed this in an elegant way and clearly shown ‘net-like’ structures by EM. Our Fig. 5D show the same thing although the sheets are more curled up and not expanded.

Genetic analysis

The manuscript reports the potential impact of two MUC5AC SNPs on MUC5AC structure and how this may have relevance for COPD and IPF, However, the analyses performed to show this are quite limited compared with recent publications assessing associations of SNPs in mucin genes with respiratory conditions (e.g., Altman et al., 2021, J Allergy Clin Immunol. Shrine et al., 2019, Lancet Respir Med). I am surprised that there is no mention of any co-variants (such as sex, age, FEV) used in the analysis. Also, the novel findings reported here need to be replicated in other cohorts. While the findings are potentially very important, my work needs to be done here.

ANSWER: We have now removed the SNP analyses in relation to disease.

Referee #4:

Sergio T. et al., report a series reconstruction of gel-forming mucins MUC5AC, and based on structural analysis combined with other approaches such as SEM and immunofluorescent staining to demonstrate the formation of net-like architecture. The authors first examined multiple structures from different constructs over MUC5AC protein, compared their structures and demonstrated how different mutations affect their structures. Then by

analyzing data from patients, authors proposed a possible role for MUC5AC in lung disease, such as IPF.

Major comments

1. In title and abstract, SNP were mentioned several times, but what's this abbreviation for? The authors should provide full name, since it's important to understand the story.

ANSWER: SNP means single nucleotide polymorphism has been explained.

2. In Fig 1D, authors described the existence of calcium ions in their model, is it sufficient to see the existence of calcium ions at a resolution around 3 angstroms? The approaches of how calcium ions were identified should be described, otherwise this figure legends as well as the figure should be changed.

ANSWER: The exact identity cannot be determined by Cryo-EM at this resolution, but other groups solved the D3 domain of MUC2 (6RBF) and VWF (6N29) by X-ray crystallography at 2.7Å and 2.5Å and they assigned calcium. Based on these structures we started using calcium for the cryoEM grids preparation. As a consequence, the quality of the particles improved drastically and we could reconstruct a map where we could see the loop and fit a calcium ion (an image generated by Coot is attached (9.00 rmsd)). In absence of calcium we could not observe any density for the entire loop, so we are confident that it binds calcium. We added a sentence explaining that the calcium ions were identified by homology.

Figure for referees not shown.

3. The information over closed-open conformation equilibrium of MUC5AC is vital to this study, yet how the 3D map on open conformation showed in Fig 3 were poorly described in Method section, and there is no information regarding how these maps were generated in Supplementary figures. Furthermore, information regarding resolution, modeling quality etc. of the open conformation maps was also not provided throughout the description in Method, neither in Supplementary figures.

ANSWER: The closed-open conformation change was initially analyzed based only on 2D classes. As we stated on the manuscript the open conformation 3D map is of low quality and has a severe preferred orientation problem. We did not give the same entity to this map as we did to the others as we considered that the quality was insufficient and it was better to stay at the 2D level. However, we realized that it was useful to illustrate the conformational change, it was easier to understand than the 2D classes for the general reader. It also allowed us to fit the different domains for a better understanding of the implications of the conformational change as they could be deduced from the 2D classes. We also noticed, when other referees asked for a ratio between closed and open conformation, that the volume is useful capturing

open conformation particles during heterologous refinement showing similar results than manual 2D classification selection but without subjective intervention. So, we ended up using the model more than intended because it was useful despite its limitations. In consequence we generated another Supplementary figure (S2) for this map and made it clear that the 3D model is a prediction based on a low-quality map.

4. Related to previous comments, in Fig 3D, authors described a putative salt bridge and a hydrogen bond, firstly, these statements should be valid only after modeling quality has been shown, secondly, distance measurement over these bonding residues should be provided on the figures, to make a better understanding over these interaction sites.

ANSWER: We made it clear that it is a tentative 3D model based on a low-quality map. We discussed putative contacts as we cannot be certain about them. The distances, 2.862 Å, 2.720 Å and 2.725 Å are not supported by real data so we prefer to keep just dashes to show that they have the potential to make the contacts.

5. In Fig. 5A authors described the tetrameric assembly of MUC5AC-D3 Arg996Gln maps, with protruding PTS domains, it is confusing whether the densities of PTS domains were actually solved in the cryo-EM map or not? If that is actual density, in Supplementary Fig 2 authors should show such density, if that is a proposed formation, authors should make a difference in showing actual density maps than PTS domain in both Fig. 5A and 5B, for better clarification.

ANSWER: Fig. 5AB are putative models of how an ideal 'net-like' structure should look like. This point was further discussed with referee #2 and #3. These figures are included to allow the reader to understand the outcome of ideal tetramerization. The text has been updated to make this clear.

6. Related to previous comment, if the formation of PTS domain was just a proposed conformation, additional information over how 40-degree angle is defined should be stated in the literature.

ANSWER: The 40-degree angle is defined as the angle generated between the vector connecting the last α carbon of chain A and chain B and the projection of the vector connecting the last α carbon of chain C and chain D in the same plane. Anyhow, we consider than a detailed description of how the angle is defined could be misleading and give the false impression that it is supported by actual data. We consider that Fig. 5A illustrates well enough the angle but we made clear in the text that it is an ideal model.

7. Clearly the statement of 'Fig S1 to S5' in Page 14, line 22 is wrong, since Fig S2 contains only local resolution estimation result, and Fig S6 also contains data processing steps. Authors should carefully check their statement about supplementary figures throughout this manuscript.

ANSWER: We have carefully checked our statements and changed the order of Supplementary figures. The old summary figure is not shown at the end as Figure 7.

8. In Supplementary Figures 1, 3, 4, 5 and 6, the particle orientation distribution exhibit preferred orientation to certain extent, 3D FSC estimation should also be supplied for these 5 maps, to indicate whether preferred orientation distribution affected map density or not.

ANSWER: Thank you for the suggestion, these has been added to all structures, now Figure 1-5.

9. In Supplementary Figures, detailed map-model fitting quality should be indicated, at least

use some representative regions such as dimerization interface to show the modeling quality over these cryo-EM maps. Also, in Supplementary Table S1, the resolution value of map-model FSC at 0.5 were provided, these FSC curves should also be supplied along with the demonstrations over map-model fitting quality.

ANSWER: We appreciate the suggestion. We included two detailed map-model fitting examples generated by Coot using different rmsd levels at two different regions, the covalent dimerization C8 area and the TIL-VWD interface to all variants. The map-model FSC curves have been also added. All to the supplemental figures 1-5.

Minor comments

1. Page 8, line 16, '*...chromosome 11 with 39,853 nucleotides in between (Figure 5F)*', lacking reference information.

ANSWER: Text deleted.

2. Page 14, line 4, authors should indicate which equipment they used to glow discharge their grids.

ANSWER: It has been added.

3. Page 14, line 8, it is unclear what the authors meant by 'dose per image'. Is it per frame or per movie?

ANSWER: It has been corrected, it is dose per frame.

4. Page 14, line 9, 'in 0.3 steps' lacking the unit of length.

ANSWER: It has been added.

5. Page 14, line 19, 'CTF correction' should be 'CTF parameters estimation' or 'defocus estimation' since correction of CTF effect doesn't happen at this step.

ANSWER: Thank you for observing this, it has been corrected.

6. Page 14, 15 and 16 lacking some essential information when describe the detailed data processing regarding each construct, such as which software was used for manual particle picking, for blob-based particle picking, for 2D classification etc. these information should be stated, and also the references should be provided.

ANSWER: We have specified that cryoSPARC was used in all the steps.

7. Naming convention should be unified throughout literature. For instance, the names on each map described in Supplementary Table are different from the ones in Supplementary Figures.

ANSWER: Thank you for observing, we have corrected typo and reviewed the text for consistency.

Recommendation

The authors provided new insight regarding how MUC5AC form the mucus. A combination of biochemistry, structural and other methods provide evidences supporting their claims, still, some of the maps and models presented in the manuscript lacking essential information, such as resolution estimation for open conformation, 3D FSC for all the reconstructed maps, modeling quality assessment etc. without these information, some of the statements made in the manuscript seem to be over claimed. Hence, I would suggest to consider this manuscript to be published after major revisions, also refinement on literatures have been done.

Dear Prof. Hansson

Thank you for the submission of your revised manuscript to EMBO Reports. We have now received the full set of referee reports that is copied below.

As you will see, both referees consider the manuscript much improved and strengthened. That said, the referees have several remaining concerns that need to be addressed. Both referees consider the model presented in Figure 5 too speculative and not supported by the data. Please address this and the other remaining referee concerns and also provide a point-by-point response.

From the editorial side there are also a number of things that need your attention:

- Please reduce the number of keywords to five.
- Reference format: et al needs to be used after 10 author names.
- On page 3 you mention unpublished data. Please either include the data to support the conclusion you describe or remove the statement on D3 dimer visualization which is based on these data.
- Your manuscript will be published in our Reports section. You have already combined the Results and Discussion section, which is fine. Reports should not exceed 27,000 characters (including spaces but excluding materials & methods and references). While this limit is not that strict, I would kindly ask you to shorten the text a bit to get closer to this number.
- The following funding information has not been entered in the online manuscript tracking system: Sahlgren's University Hospital (ALFGBG-440741). This needs to be rectified.
- Appendix: The nomenclature is Appendix Figure S#, Appendix Table S#. Please correct the Appendix and the callouts in the manuscript text.
- Please remove the paragraph
"SUPPLEMENTAL INFORMATION
Supplemental information with supplemental Figures and Table is included as an Appendix."
from the manuscript.
- Data availability section: please add URLs that resolve directly to the deposited datasets not just to the database itself.
- Source data: Please generate a folder for Figure 5, place both files into it and upload the zipped folder.
- I suggest modifying the abstract to make it more accessible to non-experts. As it stands, the term MUC5AC-D3 assembly remains unclear to the non-expert.
The following sentences could be modified to explain the specific interaction with each other (N-terminus, dimers, polymers, D domains in the N-terminus). Moreover, the term "structural single nucleotide polymorphisms" could be changed to "single nucleotide polymorphisms that affect the structure ..."
"Secreted mucins interact specifically with each other and other molecules giving mucus specific properties. We determined the cryoEM structures of the wild type MUC5AC-D3 assembly and the structural single nucleotide polymorphisms (SNP) variants R996Q and R1201W."
- Finally, EMBO Reports papers are accompanied online by
 - A) a short (1-2 sentences) summary of the findings and their significance,
 - B) 2-3 bullet points highlighting key results and
 - C) a schematic summary figure that provides a sketch of the major findings (not a data image).Please provide the summary figure as a separate file in PNG or JPG format at a size of 550x300-600 pixels (width x height). Please note that the size is rather small and that text needs to be readable at the final size. Please send us this information along with the revised manuscript.
- On a different note, I would like to alert you that EMBO Press offers a new format for a video-synopsis of work published with us, which essentially is a short, author-generated film explaining the core findings in hand drawings, and, as we believe, can be very useful to increase visibility of the work. This has proven to offer a nice opportunity for exposure i.p. for the first author(s) of the study. Please see the following link for representative examples and their integration into the article web page:
https://www.embopress.org/video_synopses
<https://www.embopress.org/doi/full/10.15252/emj.2019103932>

Kind regards,

Referee #1:

The manuscript has been greatly improved by removal of the MUC5AC SNP data and its potential association with lung disease. However, there are some remaining concerns that should be addressed.

The authors have not addressed why the MUC5AC-N covalent dimer structure is shown in the reduced form (which is claimed to be a partial variant of MUC5AC-N). Why is the structure of the oxidised form not presented shown?

The schematic representation of the ideal structure of the MUC5AC non-covalently cross-linked network lacks the supporting evidence for (1) the regularity of network crosslinks, and (2) that "the PTS from each covalent dimer extend in opposite directions forming an angle of about 40{degree sign} with the PTS from the other dimer." Fig 5A and 5B are mainly speculative and should be removed.

Referee #2:

Sergio T. et al., reported several reconstructions of mucins MUC5AC and its variants. Based on structural analysis combined with other approaches, the authors proposed a model of how the gel-forming structure of MUC5AC. After previous revision, the authors clearly improved the manuscript a lot, most of my previous concerns were addressed. and I find this to be an interesting manuscript to read and could potentially draw the broader interest of the community. Still, the structural analysis would be more solid and I suggest this manuscript to be considered to publish, after the following points has been addressed.

1. In Fig 3D, even after the revision, the putative bonds are presented in a way that is very hard to distinguish from the covalent bonds. Also, since the authors proposed 'potential contact', I am strongly against to put such 'dashed line' in main figures, could the authors move this to supplementary figures to make a less strong statement, or just delete those lines from the main figure?
2. I'm still not satisfied with Fig 5A & 5B, firstly, these two provide redundant information, so I suggest the authors delete one of the two. Secondly, this image presents a putative model of how MUC5AC network forms, and I find it very misleading when then authors combined the schematic view with actual model in the same figure, furthermore, it is not something observed from experimental data, so I would suggest the authors to change this figure to schematic view, without any model.

EMBO Reports

January 3, 2025

Editor Martina Rembold

Dear Martina,

Thank you for your positive interest in this manuscript. We have now addressed all points and I hope to your satisfaction as stated in the point-by-point answers to the Editorial and Referee comments.

We have remade the Fig. 5A-B to make them appear less specific, but we still think that these or similar cartoon are required for the non-expert reader to understand the points of the manuscript.

We now hope that you can find the submission acceptable for publication.

Happy New Year!!

Gunnar C. Hansson, MD, PhD

Professor

Editorial and Referee Comments and Answers

- Please reduce the number of keywords to five.

ANSWER: Done

- Reference format: et al needs to be used after 10 author names.

ANSWER: Corrected

- On page 3 you mention unpublished data. Please either include the data to support the conclusion you describe or remove the statement on D3 dimer visualization which is based on these data.

ANSWER: The statement has been deleted as the data referred to are included later on in the text and thus there was not any unpublished data.

- Your manuscript will be published in our Reports section. You have already combined the Results and Discussion section, which is fine. Reports should not exceed 27,000 characters (including spaces but excluding materials & methods and references). While this limit is not that strict, I would kindly ask you to shorten the text a bit to get closer to this number.

ANSWER: We have deleted some text belonging to the introduction and discussed parts. We did not want to delete anything from the results text as this might cause additional

problems with the referees. We have cut down, but are still slightly longer, something we hope is acceptable.

- The following funding information has not been entered in the online manuscript tracking system: Sahlgren's University Hospital (ALFGBG-440741). This needs to be rectified.

ANSWER: Has been corrected in the submission system.

- Appendix: The nomenclature is Appendix Figure S#, Appendix Table S#. Please correct the Appendix and the callouts in the manuscript text.

ANSWER: Has been corrected in Supplement and Methods and controlled in the other text.

- Please remove the paragraph

"SUPPLEMENTAL INFORMATION

Supplemental information with supplemental Figures and Table is included as an Appendix." from the manuscript.

ANSWER: The whole paragraph has been removed.

- Data availability section: please add URLs that resolve directly to the deposited datasets not just to the database itself.

ANSWER: We have added these and should work once the pdb data has been released.

- Source data: Please generate a folder for Figure 5, place both files into it and upload the zipped folder.

ANSWER: Done and submitted

- I suggest modifying the abstract to make it more accessible to non-experts. As it stands, the term MUC5AC-D3 assembly remains unclear to the non-expert.

The following sentences could be modified to explain the specific interaction with each other (N-terminus, dimers, polymers, D domains in the N-terminus). Moreover, the term "structural single nucleotide polymorphisms" could be changed to 'single nucleotide polymorphisms that affect the structure

"Secreted mucins interact specifically with each other and other molecules giving mucus specific properties. We determined the cryoEM structures of the wild type MUC5AC-D3 assembly and the structural single nucleotide polymorphisms (SNP) variants R996Q and R1201W."

ANSWER: We have altered as suggested with some modifications.

- Finally, EMBO Reports papers are accompanied online by

A) a short (1-2 sentences) summary of the findings and their significance,

ANSWER: Two sentences added on page 1

B) 2-3 bullet points highlighting key results and

ANSWER: Two bullet points added on page 1

C) a schematic summary figure that provides a sketch of the major findings (not a data image).

Please provide the summary figure as a separate file in PNG or JPG format at a size of

550x300-600 pixels (width x height). Please note that the size is rather small and that text needs to be readable at the final size. Please send us this information along with the revised manuscript.

ANSWER: Schematic figure (550x281 pixel) has been made and included

- On a different note, I would like to alert you that EMBO Press offers a new format for a video-synopsis of work published with us, which essentially is a short, author-generated film explaining the core findings in hand drawings, and, as we believe, can be very useful to increase visibility of the work. This has proven to offer a nice opportunity for exposure i.p. for the first author(s) of the study. Please see the following link for representative examples and their integration into the article web page:

<https://www.embopress.org/doi/full/10.15252/emboj.2019103932>

ANSWER: We think this is an excellent initiative that should be especially useful when there is videos included in the paper. We have experienced that the readers are not down loading and looking at videos as these are often saying much more that static figures in dynamic processes. However, here we have structural information and to videos and thus we have decided not to generate one.

Referee #1:

The manuscript has been greatly improved by removal of the MUC5AC SNP data and its potential association with lung disease. However, there are some remaining concerns that should be addressed.

ANSWER: Thank you for the positive comment.

#1: The authors have not addressed why the MUC5AC-N covalent dimer structure is shown in the reduced form (which is claimed to be a partial variant of MUC5AC-N). Why is the structure of the oxidised form not presented shown?

ANSWER: The reported structure is the most abundant that we have observed. This has been quite extensively discussed in the previous version and we do not have more to add to this discussion. The structure presented is the most frequent one, the way we think that data should be presented. That this reduced disulfide was also found previously in MUC2 as discussed, suggest that its presence might be important. We had unfortunately omitted the reference here, but the reference was presented several times. In this article Javitt et al attributed the presence of the reduced C-C bond to X-ray radiation damage, but as we also observed this although our Cryo-EM analyses could of course also had cause radiation damage. We believe that there could be a reason for the presence of a labile reduced linkage and have pointed out a structural explanation and discussed a putative biological implication. We claimed that the reduction is partial because by adjusting the cryoEM map threshold it is possible to see some density between the cysteines. However, due to the resolution limitation and to prevent overinterpretation of the maps, we decided not to model alternative conformations with low frequency. We believe that the presented model and text most correctly reflect observed results and that this could have biological importance.

We would like to keep the present text on Page 8 last paragraph that reads (missing reference added): Interestingly no major structural differences were observed compared to the low pH. It indicates that once the covalent N-terminal dimerization is stabilized the interface is locked, even if the Cys1132-Cys1132 dimer bond is partially reduced. This disulfide bond reduction was first observed for the equivalent Cys observed in the MUC2 crystallographic structure (Javitt et al., 2020). This was then attributed to radiation damage, an event that could also occur in cryoEM and can provide information about "weak links" that can be of structural significance (Weik et al., 2000). This suggests that the formation of the Cys1132-Cys1132 disulfide bond could be tightly regulated by its redox potential, requiring the higher oxidizing environment found in the Golgi compared to the ER (Kellokumpu, 2019). Its higher radiation damage susceptibility could also be explained by the formation of a stabilizing S...O interaction (Bhattacharyya et al., 2020) with the carbonyl O-atom from the same cysteine located just at 3.3 Å from the S-atom.

#1:The schematic representation of the ideal structure of the MUC5AC non-covalently cross-linked network lacks the supporting evidence for (1) the regularity of network crosslinks, and (2) that "the PTS from each covalent dimer extend in opposite directions forming an angle of about 40{degree sign} with the PTS from the other dimer." Fig 5A and 5B are mainly speculative and should be removed.

ANSWER: We agree that the Fig. 5A and B were easily interpreted as definite and not models as written in the legend. However, we believe that the readers that know less about mucins need to have some guidance into how the TIL-D3 interactions will form net-like structures. We have redrawn the Fig. 5A to point out where the four PTS domains start and

given them variable, possible directions. We have also remade the Fig. 5B to show the how long linear MUC5AC filaments will form one D3 interaction shown as an example. By the Fig. 5A and B, the non-expert reader should be able to understand how net-like structures can be formed. See also below.

Referee #2:

Sergio T. et al., reported several reconstructions of mucins MUC5AC and its variants. Based on structural analysis combined with other approaches, the authors proposed a model of how the gel-forming structure of MUC5AC. After previous revision, the authors clearly improved the manuscript a lot, most of my previous concerns were addressed. and I find this to be an interesting manuscript to read and could potentially draw the broader interest of the community. Still, the structural analysis would be more solid and I suggest this manuscript to be considered to publish, after the following points has been addressed.

ANSWER: Thanks for the positive comments.

1. In Fig 3D, even after the revision, the putative bonds are presented in a way that is very hard to distinguish from the covalent bonds. Also, since the authors proposed 'potential contact', I am strongly against to put such 'dashed line' in main figures, could the authors move this to supplementary figures to make a less strong statement, or just delete those lines from the main figure?

ANSWER: We agree and have deleted the dashed line as suggested and alter the legend accordingly.

2. I'm still not satisfied with Fig 5A & 5B, firstly, these two provide redundant information, so I suggest the authors delete one of the two. Secondly, this image presents a putative model of how MUC5AC network forms, and I find it very misleading when then authors combined the schematic view with actual model in the same figure, furthermore, it is not something observed from experimental data, so I would suggest the authors to change this figure to schematic view, without any model.

ANSWER: We have altered the Fig. 5A to include only arrows showing where the remaining C-terminal part of the four MUC5ACs start and by multiple arrow that these can have are flexible directions. For the 5B we have drawn two MUC5AC covalent filaments and how these extend as implied by the arrows. We give one example to show one of these the D3-TIL interactions. We hope that this will help especially non-expert readers as discussed for referee #1.

Prof. Gunnar Hansson
University of Gothenburg
Dept. of Medical Biochemistry
Medicinaregatan 9C
Gothenburg, VG 40530
Sweden

Dear Gunnar,

Thank you for the submission of your revised manuscript and please apologize my delayed response, which is caused by a backlog that accumulated over the Christmas/New Year period. That said, I have now gone through the revised manuscript and am very pleased to accept it for publication in the next available issue of EMBO reports. Thank you for your contribution to our journal.

Kind regards,

Martina
